# VIDEOPHY-2: A CHALLENGING ACTION-CENTRIC PHYSICAL COMMONSENSE EVALUATION IN VIDEO GENERATION

**Hritik Bansal**[*1]  **Clark Peng**[*1]  **Yonatan Bitton**[*2]
**Roman Goldenberg**[2]  **Aditya Grover**[1]  **Kai-Wei Chang**[1]

[1]**University of California Los Angeles**  [2]**Google Research**

## ABSTRACT

Large-scale video generative models, capable of creating realistic videos of diverse visual concepts, are strong candidates for general-purpose physical world simulators. However, their adherence to physical commonsense across real-world actions remains unclear (e.g., playing tennis, backflip). Existing benchmarks suffer from limitations such as limited size, lack of human evaluation, sim-to-real gaps, and absence of fine-grained physical rule analysis. To address this, we introduce VIDEOPHY-2, an action-centric dataset for evaluating physical commonsense in generated videos. We curate 4000 diverse and detailed prompts for video synthesis from modern generative models. We perform human evaluation that assesses semantic adherence, physical commonsense, and grounding of physical rules in the generated videos. Our findings reveal major shortcomings, with even the best model achieving only 47.7% joint performance (i.e., high semantic and physical commonsense adherence) on the hard subset of VIDEOPHY-2. We find that the models particularly struggle with conservation laws like mass and momentum. Finally, we also develop VIDEOPHY-2-AUTOEVAL, an automatic evaluator for fast, reliable assessment on our dataset. Overall, VIDEOPHY-2 serves as a rigorous benchmark, exposing critical gaps in video generative models and guiding future research in physically-grounded video generation. The data and code is available at https://videophy2.github.io/.

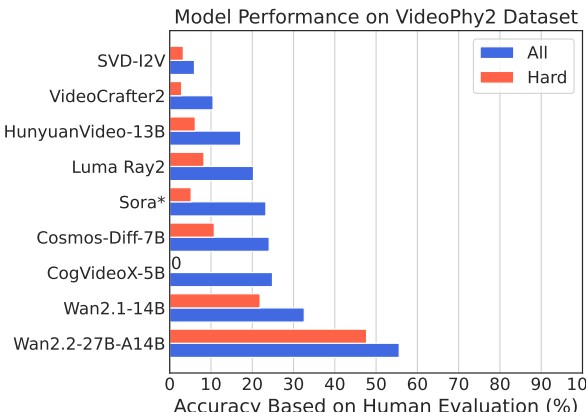

Figure 1: **Performance on the VIDEOPHY-2 dataset using human evaluation.** We evaluate the physical commonsense and semantic adherence to text conditioning prompts for diverse real-world actions. We observe that even the best-performing model Wan2.2-27B-A14B (27B total, 14B active params) achieves 47.7% on the hard subset of the data, created using CogVideoX-5B as a reference model. * represents the evaluation on a small subset of the dataset.

## 1 INTRODUCTION

Recent advancements in large-scale video generative modeling offer the potential to simulate the physical world accurately [13, 58]. In particular, this capability can enable learning general-purpose visuomotor policies [36, 19], autonomous driving [1], and game playing [15, 21, 5]. In daily life, humans rely on their sophisticated physics intuition to interact with the world [20] (e.g., predicting

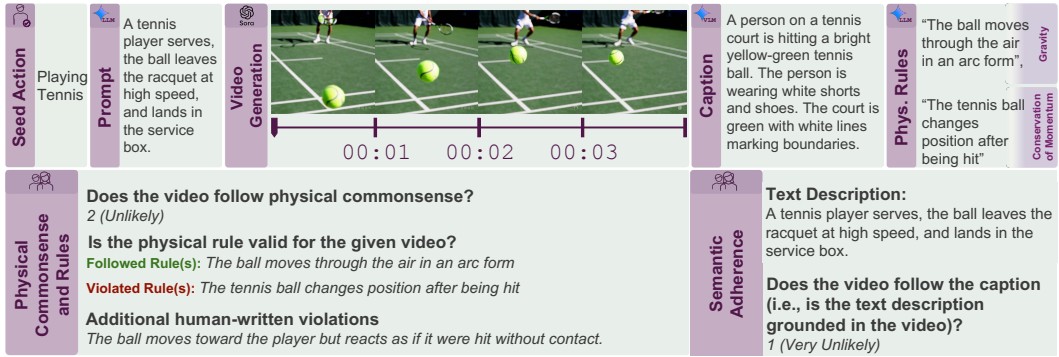

Figure 2: **VIDEOPHY-2 pipeline.** We generate a text prompt from the seed action using an LLM, create a video with a video model, and caption it with a VLM to extract candidate physical rules. Then, humans rate the video's physical likelihood, verify rule violations, suggest missing rules, and assess prompt adherence.

the trajectory of football after being hit). However, the extent to which existing video models can generate physically likely worlds across diverse real-world actions remains unclear.

An approach to evaluating generated videos is to compare them with ground-truth physical simulations [1, 52]. Furthermore, there is a lack of mature methods for rendering diverse real-world materials [8, 31, 49] and for accurately simulating complex physical interactions [40]. For instance, simulating a scenario like 'a child kicking a ball against a wall' requires precise estimation of the foot's pose, and considerations of the ball's air pressure and material properties. While we focus on evaluating the physical likelihood of generated videos, an assessment that can often be made by humans without formal physics education by relying on their real-world experience.

Recent work such as Physics-IQ [48] conditions video models on the first few frames of real videos and evaluates their similarity by comparing predicted videos with ground-truth completions. However, this approach faces several challenges: (a) the extent to which it agrees with human judgment remains unclear, and (b) extending it to more complex scenarios depicting multiple events is non-trivial. Another work PhyGenBench [40] curates a small set of 160 manually crafted prompts, which is not scalable. Additionally, their evaluation approach simplifies the problem by designing text prompts that explicitly associate with a single physical law (e.g., 'A stone placed on the surface of a water pool' is linked to law of Buoyancy). Although, this strict one-to-one association between a prompt and a physical law is problematic, as video models often exhibit imperfect semantic adherence. For instance, a video model might generate a video that does not strictly follow the prompt but still adheres to physical commonsense (e.g., producing a video where 'a stone is dropped from a height into the pool', where gravity is more crucial than buoyancy). We note the difference between VIDEOPHY-2 and several existing work in Table 1.

To address these gaps, we propose VIDEOPHY-2, a challenging physical commonsense evaluation dataset for real-world actions. Specifically, we curate a list of **197 actions** across diverse physical activities (e.g., hula-hooping, gymnastics) and object interactions (e.g., bending an object until it breaks). Then, we generate 3940 detailed prompts from these seed actions using a large language model (LLM). Further, these prompts are used to synthesize videos with modern video generative models. Finally, we compile a list of **candidate physical rules** (and laws) that should be satisfied in the generated videos, using vision-language models in the loop. For example, in a video of *sportsperson playing tennis*, a physical rule would be that *ball should follow a parabolic trajectory under gravity*. For gold-standard judgments, we ask human annotators to score each video based on semantic adherence and physical commonsense, and to mark its compliance with physical rules.

In our experiments, we find that the best-performing model, Wan2.2-27B-A14B [60], achieves a joint performance score (high semantic adherence and physical commonsense) of only $55.4\%$ while Wan2.1-14B achieves a score of $32.6\%$. To further increase the dataset's difficulty, we create a **hard subset** where the performance of Wan2.2-27B-A14B and Wan2.1-14B drops to $47.7\%$ and $22\%$, respectively. Moreover, our fine-grained analysis of human-annotated physical rule violations shows that video models struggle the most with *conservation laws* (e.g., mass and momentum).

While human evaluation serves as the gold standard for real-world physical commonsense judgment, it is expensive and difficult to scale. To address this, we train an automatic evaluation model,

Table 1: **Comparison between VIDEOPHY-2 and several prior work.** We highlight the salient features of the VIDEOPHY-2 and show that its unique contributions. For instance, it is one of the largest datasets for physical commonsense evaluation, along with violated physical rule (and law) annotations.

| Feature | VBench[28] | PhyGenBench [47] | PhysicsIQ [48] | EvalCrafter [41] | VIDEOPHY [8] | VIDEOPHY-2 (Ours) |
|---|---|---|---|---|---|---|
| Num of captions | 1746 | 160 | 396 | 700 | 688 | 3940 |
| Gold human evaluation | ✓ | ✓ | ✗ | ✓ | ✓ | ✓ |
| Physical commonsense eval. | ✗ | ✓ | ✓ | ✗ | ✓ | ✓ |
| Physical rules and laws annotations | ✗ | ✓ | ✗ | ✗ | ✗ | ✓ |
| Real-world action-centric | ✗ | ✗ | ✗ | ✗ | ✗ | ✓ |
| Long (dense) captions | ✗ | ✓ | ✓ | ✗ | ✗ | ✓ |
| Hard subset | ✗ | ✗ | ✗ | ✗ | ✗ | ✓ |
| Automatic evaluator | ✓ | ✓ | ✓ | ✓ | ✓ | ✓ |
| Release videos and annotations | ✓ | ✗ | ✗ | ✓ | ✓ | ✓ |
| Human feedback type | Pairwise | Rating | - | Rating (1-5) | Rating (0-1) | Rating (1-5) |

**VIDEOPHY-2-AUTOEVAL**, capable of performing a wide range of tasks—including scoring semantic adherence, physical commonsense, and classifying physical rule grounding in the generated video. In our experiments, we find that VIDEOPHY-2-AUTOEVAL outperforms a capable multimodal foundation model, Gemini-2.0-Flash-Exp [18], with a relative correlation improvement of $81\%$ and $236\%$ on the semantic adherence and physical commonsense tasks, respectively, on the unseen prompts. Overall, we demonstrate that VIDEOPHY-2 is a high-quality dataset that poses a formidable challenge for modern video models.

## 2 VIDEOPHY-2 DATASET

We present the steps for data construction (Figure 2) below:

**Seed Actions (Stage 1):** We curate a set of actions relevant to physical commonsense evaluation. Specifically, we compile a diverse list of over 600 actions from popular video datasets that capture a wide range of real-world activities, particularly those involving sports, physical activities, and object interactions. These datasets include Kinetics [16], UCF-101 [55], and SSv2 [22]. Next, we divide the student authors, with undergraduate or more degree in STEM, into two groups, each of which independently reviews the list and marks actions deemed relevant for physical commonsense evaluation. Our goal is to include actions that test various physical laws (e.g., gravity, elasticity, buoyancy, reflection, conservation of mass and momentum). Importantly, we filter out actions that do not elicit significant motion or are unlikely to be compelling for physical commonsense evaluation in videos (e.g., typing, applying cream, arguing, auctioning, chewing, playing instruments, petting a cat). Finally, we retain only the actions deemed relevant by both groups of annotators. After this filtering process, we obtain a list of 232 actions, which we further refine using Gemini-2.0-Flash-Exp to remove semantic duplicates, resulting in a final set of **197** actions. Among these, 143 and 54 actions belong to object interactions and physical activities category, respectively. We present the list in Appendix Table 9.

**LLM-Generated Prompts (Stage 2):** In this stage, we query the Gemini-2.0-Flash-Exp LLM to independently generate 20 prompts for each action in our dataset. In particular, we focus on the depiction of multiple events within a prompt to increase the challenge for modern video generation models (e.g., we encourage the LLM to generate 'An archer draws the bowstring back to full tension, then releases the arrow, which flies straight and strikes a bullseye on a paper target' instead of a simpler prompt 'An archer releases the arrow'). Our prompt generation template is presented in Appendix F. In total, we curate **3940** prompts covering a wide range of actions. Since the modern video models can understand long video descriptions, we also generate dense captions from the original captions using the Mistral-NeMo-12B-Instruct prompt upsampler from [1]. In particular, these dense captions add more visual details to the original caption without changing its semantic meaning (e.g., main characters and actions).[1]

**Candidate physical rules and laws (Stage 3):** In this stage, we aim to generate candidate physical rules and associated laws that could be followed (or violated) in the generated video. Since video models often struggle to adhere to conditioning text prompts, we do not derive physical rules directly from them. Instead, we first generate videos using generative models conditioned on prompts from

---

[1]We present some of the generated captions and underlying actions in Appendix Table 13, and the upsampled captions in Appendix Table 14.

the VIDEOPHY-2 dataset. Then, we create captions for these videos using the strong video captioning capabilities of Gemini-2.0-Flash-Exp. This ensures that the physical rules are constructed based on details grounded in the video itself.[2] Subsequently, we ask Gemini-2.0-Flash-Exp to generate a set of three physical rules (and laws) that should be followed for a given video. Since a video may violate physical rules that are not covered in the pre-defined rules, we further ask the human annotators to write additional violated rules during physical commonsense evaluation. We present the rule generation prompt in Appendix Table 11.

**Construction of the Hard Subset (Stage 4):** While we collect diverse and lengthy captions to make the task more challenging, we further employ a model-based strategy to identify a subset of particularly difficult actions. Specifically, we generate videos using a strong open video model, CogVideoX-5B [63], conditioned on captions from the VIDEOPHY-2 dataset. From this, we select 60 actions (out of 197) for which the model fails to generate videos that accurately adhere to the prompts and follow physical commonsense (Appendix Table 10). On examination, we find that these actions focus on physics-rich interactions (e.g., momentum transfer in throwing discus or passing football), state changes (e.g., bending something until it breaks), balancing (e.g., tightrope walking), and complex motions (e.g., backflip, pole vault, and pizzatossing). In total, we designate 1200 prompts making the dataset more challenging. We present the list of hard actions in Appendix Table 10.[3]

**Data Analysis:** We present the dataset statistics in Appendix Table 7. Specifically, VIDEOPHY-2 contains 3940 captions, which is $5.72\times$ more than those in the VIDEOPHY dataset. Additionally, the average lengths of original and upsampled captions are 16 and 138 tokens, respectively—$1.88\times$ and $16.2\times$ longer than those in VIDEOPHY. Furthermore, VIDEOPHY-2 includes 110K human annotations across various video generative models and their semantic adherence, physical commonsense, and physical rule annotations. Finally, we show the distribution of the root verbs and direct nouns in the original captions of VIDEOPHY-2 in Appendix Figure 6, demonstrating the high diversity of the dataset. We also illustrate the diversity of multiple captions for an action in Appendix Figure 7.

## 3 EVALUATION

### 3.1 METRIC

In practice, generated videos must satisfy several constraints, including high video quality [41], temporal consistency [28], and entity/background consistency [7]. While many of these metrics are intertwined, it is essential to evaluate each independently to gain a clearer understanding of a model's capabilities. In this work, we focus on assessing the extent to which a generated video (1) *adheres to the input text prompt* and (2) *follows physical commonsense*. To quantify these aspects, we employ a rating-based evaluation using a 5-point Likert scale, a well-established methodology for capturing human judgments across domains ranging from psychology [37, 2] to large language model evaluation [50]. This approach has also been adopted in video model evaluation [8, 40, 41]. Unlike ranking-based feedback, which only reflects relative preferences between two outputs, our rating system measures the absolute degree of a model's success or failure. Moreover, the 5-point scale provides more fine-grained feedback than binary labels (e.g., plausible/implausible), enabling a more nuanced analysis of model performance.

Since human evaluation inherently involves subjectivity, we implemented a rigorous protocol to ensure reliability. All annotators underwent structured training guided by a detailed rubric with clear examples, ranging from "very unlikely" (1) to "very likely" (5), to anchor their judgments and establish a shared understanding of the scale. To further reduce individual bias and capture a stable consensus, each video was evaluated by three annotators. This process yielded a high inter-annotator agreement of $80\%$ (comparable to agreement scores reported in prior work [8]), confirming the consistency and validity of our framework.

---

[2]We observe that prompting Gemini-2.0-Flash-Exp to generate physical rules directly from the video did not yield high-quality outputs. Therefore, we prefer a two-step process: captioning followed by rule generation.

[3]We note that a similar model-based strategy is also adopted in recent work like Humanity's Last Exam [51] and ZeroBench [54] to collect hard instances for model evaluation.

**Semantic Adherence (SA):** Here, we aim to assess whether the input text prompt is semantically grounded in the generated video. Specifically, it studies whether the entities, actions, and relationships described in the prompt are accurately depicted in the video (e.g., a person visibly jumping into the water). To measure semantic adherence, annotators rate each video on a 5-point scale, selecting from the following options: $\{SA \in$ *Very Unlikely (1), Unlikely (2), Neutral (3), Likely (4), Very Likely (5)*$\}$. In this case, *very unlikely* indicates that the video does not match the prompt at all, and *very likely* highlights the video fully adheres to the prompt with no inconsistencies.

**Physical Commonsense (PC):** Here, our goal is to assess whether the generated video follows the physical laws of the real-world intuitively (e.g., the football should start moving after impact in accordance with newton's first law). We note that the physical commonsense evaluation is independent of the underlying video generating text prompt. Since a video can follow (or violate) numerous laws, we are concerned with the holistic sense of the video's physical commonsense. In particular, the annotators rate each video on a 5-point scale, selecting from the following options: $\{PC \in$ *Very Unlikely (1), Unlikely (2), Neutral (3), Likely (4), Very Likely (5)*$\}$. Here, *very unlikely* that the video contains numerous violations of fundamental physical laws, and *very likely* indicates that the video demonstrates a strong understanding of physical commonsense with no violations.

Similar to [8], we compute **joint performance** as the main evaluation metric, which measures the fraction of videos that both adhere closely to the text prompt ($SA \geq 4$) and follow physical commonsense to a high degree ($PC \geq 4$). We do not report the posterior score ($PC >= 4|SA >= 4$) since a bad model can game it.[4]

**Physical Rules (PR):** A key feature of the VIDEOPHY-2 dataset is the collection of candidate physical rules (and their associated laws) that humans evaluate as being followed or violated in the generated video (e.g., 'the ball should go down' is a physical rule associated with the law of gravity). These rules enable a fine-grained assessment of the video model's capabilities. Specifically, we determine whether a candidate physical rule is *violated (0), followed (1)*, or *cannot be determined (CBD) (2)* in the generated video.[5] Further, we ask human annotators to note more physical rule violations to ensure comprehensive coverage.

## 3.2 SCORING

**Human Evaluation.** In practice, human evaluation serves as a gold standard for assessing the quality of generative foundation models [43, 64]. In particular, we collect judgments using the Amazon Mechanical Turk (AMT) platform from a group of 12 human annotators, which were selected after passing a qualification test. Since physical commonsense is independent of the generated video-prompt alignment, we evaluate semantic adherence and physical commonsense (including rule-based judgment) as separate tasks for human annotators. This differs from prior work in VIDEOPHY [8], which treats semantic adherence and physical commonsense assessment as a single task. It may introduce evaluation bias, as annotators have access to the prompt while conducting the physical commonsense evaluation, a scenario we explicitly avoid in this work.

We present the annotation UI for the semantic adherence task in Appendix Figure 17, where the input consists of a text prompt and the corresponding generated video. Note that human annotators were shown the original prompt (not the upsampled prompts) to ensure a fair comparison between video models, regardless of their ability to handle short or long prompts. In the following task, human annotators are asked to evaluate only the generated video and with regard to adherence to specific physical rules (followed/violated/CBD), overall physical commonsense, and observable behaviors that violate physical reality.[6] The annotation interface is shown in Appendix Figure 18.

**Automatic Evaluation.** While human judgments serve as the gold standard, automating the evaluation process is crucial for faster and more cost-effective model assessments. In this study, we evaluate

---

[4]A video model can adhere to the prompt for only 1 out of 1,000 prompts in the dataset. Now, assume that this video is also physically realistic. In this case, the posterior performance of the model will be reported as 100%, which is quite misleading for the model builders.

[5]We include CBD category because LLM-generated physical rules may not be grounded in the video.

[6]In our instructions to the annotators, we explicitly clarify that the overall physical commonsense judgments should extend beyond the predefined physical rules listed in the task.

several video-language foundation models (e.g., Gemini-2.0-Flash-Exp, VideoScore [23]) on two tasks: semantic adherence and physical commonsense scoring. Specifically, we prompt the models to score generated videos based on these two criteria and then normalize their predictions to a 5-point scale. We provide more details about score computation in Appendix K. Additionally, we introduce a classification task to determine whether a given physical rule is followed, violated, or CBD in the generated video, leveraging video-language models such as VideoLLaVA [38]. Here, we prompt the model to classify each video-rule pair into one of three categories: followed, violated, or CBD.

## 4 SETUP

**Video generative models.**    In this work, we evaluate a diverse range of state-of-the-art text-to-video generative models. Specifically, we assess seven open models and two closed models, including *CogVideoX-5B* [63], *VideoCrafter2* [17], *HunyuanVideo-13B* [34], *Cosmos-Diffusion-7B* [1], *Stable Video Diffusion (SVD-I2V)* [11], *Wan2.1-14B* [60], *Wan2.2-T2V-27B-A14B* (MoE with 27B total and 14B active params) [60], *OpenAI Sora* [13], and *Luma Ray2* [44].[7] We prompt these models with the upsampled captions, except for those that do not support long (dense) captions by design i,e., Hunyuan-13B and VideoCrafter2. For SVD-I2V, we first generate an image using Stable Diffusion and then use it as a conditioning variable to SVD. Additionally, we generate short videos (less than 6s) as they are easier to evaluate and effectively highlight challenges on the VIDEOPHY-2. The model inference details are provided in Appendix M.

**Dataset setup.**    Similar to [8], we take a data-driven approach and use human annotations across multiple tasks to train the automatic evaluator. We split the VIDEOPHY-2 dataset into a test set for benchmarking and a training set for training the VIDEOPHY-2-AUTOEVAL model. Specifically, the training and testing prompts consist of 3350 (197 actions × 17 captions per action) and 590 (197 actions × 3 captions per action) prompts, respectively.

**Benchmarking.** For every tested model, we generate one video per each test prompt, that is, 591 videos per model.[8]  After generating the videos, we ask three annotators to evaluate them based on semantic adherence, overall physical commonsense, and violations of various physical rules. Annotators can also suggest additional physical rules that may be missing from our list. For every generated video, we compute the SA and PC scores (1-5) by averaging the three annotators scores and rounding to the nearest integer. Following this, the joint score is computed to assess the quality of the generated video. We use the majority voting for determining whether the listed physical rule (and law) is followed, violated, or cannot be grounded in the generated video. Additional human-written violations are converted to a statement of a physical rule (and law) using Gemini-2.0-Flash-Exp. With CogVideoX-5B as a strong reference model, we choose a *hard* subset of 60 actions for which it achieved a zero joint performance. In our experiments, we observe that this hard subset leads to big drop in performances in comparison to the entire data across video models.

**AutoEvaluation.** Our experiments reveal that existing video-language models struggle to achieve strong agreement with human annotators. This discrepancy primarily arises due to their limited understanding of physical commonsense and rules, as well as the complexity of the prompts. Hence, we supplement our benchmark with a video-language model VIDEOPHY-2-AUTOEVAL (7B parameters). Specifically, we aim to provide more accurate predictions for the generated videos along three axis – semantic adherence score (1-5), physical commonsense score (1-5), and physical rule classification (0-2). We follow a data-driven approach to distill human knowledge into a foundation model for these tasks. Specifically, we fine-tune a video-language model VideoCon-Physics [8] on 50K human annotations acquired for these tasks. We train a mult-task model to solve the three tasks using a shared backbone, to allow the inter-task knowledge transfer. We provide the templates and setup used for model finetuning in Appendix J and Appendix I, respectively.

**Training set for VIDEOPHY-2-AUTOEVAL.**    Within a limited data collection budget, we sample 1 video per caption from one of the three video models including HunyuanVideo-13B, Cosmos-

---

[7]We exclude other closed models due to lack of API access (e.g., Veo2 [58], Kling [32]).

[8]For Sora, however, we generate a subset of 60 videos (randomly selected from 591), manually, using Sora playground (https://openai.com/sora/) due to the lack of an official API, and 394 videos (2 prompts per action) for Ray2 due to the limited API budget.

Table 2: **Human evaluation results on VIDEOPHY-2.** We present the joint performance that focuses on high semantic adherence and high physical commonsense in the generated videos. PA, OI refer to the physical activities, and object interactions subsets of the data, respectively. We mark the best performing models in each column by blue and second best by yellow.

| Model | Class | All | Hard | PA | OI |
|---|---|---|---|---|---|
| Wan2.2-27B-A14B [60] | Open | 55.4 | 47.7 | 54.5 | 58.6 |
| Wan2.1-14B [60] | Open | 32.6 | 21.9 | 31.5 | 36.2 |
| CogVideoX-5B [63] | Open | 25.0 | 0.0 | 24.6 | 26.1 |
| Cosmos-Diff-7B [1] | Open | 24.1 | 10.9 | 22.6 | 27.4 |
| Hunyuan-13B [34] | Open | 17.2 | 6.2 | 17.6 | 15.9 |
| VideoCrafter-2 [17] | Open | 10.5 | 2.9 | 10.1 | 13.1 |
| SVD-I2V [17] | Open | 6.0 | 3.3 | 5.2 | 8.7 |
| Ray2 [44] | Closed | 20.3 | 8.3 | 21.0 | 18.5 |
| Sora [13] | Closed | 23.3 | 5.3 | 22.2 | 26.7 |

Diffusion-7B, and CogVideoX-5B from the training set, of size 3350. All other video models are used to study the generalization capabilities of the auto-rater. Subsequently, we perform human annotations in the same way as the benchmarking process i.e., aggregating semantic adherence, physical commonsense and rule judgments across the three annotators. In total, we collect roughly 50K human annotations across the three tasks, and spend $3515 USD on collecting the training data. Post-training, we compare the performance of VIDEOPHY-2-AUTOEVAL against several baselines on the semantic adherence and physical commonsense judgments using Pearson's correlation between the ground-truth and predicted scores. Further, we compare the joint score prediction accuracy and F1 score between our auto-rater and selected baselines. In addition, we compare the physical rule classification accuracy between the VIDEOPHY-2-AUTOEVAL and baselines.

## 5 EXPERIMENTS

Here, we present the benchmarking results and the fine-grained analysis (§5.1). Then, we note the usefulness of our auto-rater against modern video-language models (§5.2).

### 5.1 MAIN RESULTS

**Performance on the dataset.** We compare the joint performance of various open and closed text-to-video generative models on the VIDEOPHY-2 dataset in Table 2. Specifically, we present their performance on the entire dataset, the hard split, and subsets focused on physical activities/sports (PA) and object interactions (OI). Even the best-performing model, Wan2.2-27B-A14B, achieves 55.4% and 47.7% (14% relative reduction) on the full and hard splits of our dataset, respectively. On the other hand, we find that the second-best model, Wan2.1-14B achieves only 32.6% and 21.9% (33% relative reduction) on the full and hard splits, respectively. This highlights at the effectiveness of higher model capacity in Wan2.2 in comparison to Wan2.1 without increase in inference cost using mixture-of-experts for different denoising timesteps. Furthermore, we observe that closed models, such as Ray2, perform worse than open models like Wan2.2-27B-A14B and CogVideoX-5B. This suggests that closed models are not necessarily superior to open models in capturing physical commonsense.

Additionally, we find that performance on physical activities (sports) is generally lower than on object interactions across different video models. This suggests that future data curation efforts should focus on collecting high-quality sports activity videos (e.g., tennis, discus throw, baseball, cricket) to improve performance on the VIDEOPHY-2 dataset. Finally, we present the correlation between SA and PC judgments and other video metrics, including aesthetics (measured using the LAION classifier [35]) and motion quality (measured using optical flow from RAFT [56]), in Table 3. Our results reveal that phys-

Table 3: **Correlation analysis between semantic adherence and physical commonsense with aesthetics and motion video metrics.**

| | Aesthetics | Motion | SA |
|---|---|---|---|
| SA | 0.1 | 0.02 | 1 |
| PC | 0.09 | 0.002 | 0.14 |

ical commonsense is not well-correlated with any of these video metrics. This indicates that a model cannot achieve high performance on our dataset simply by optimizing for aesthetics and motion qual-

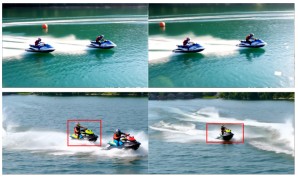 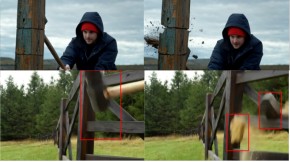 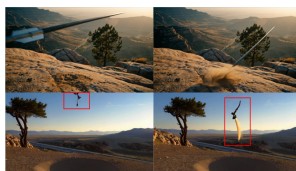

(a) **v. Ray2**, Two jetskis race side-by-side, their wakes colliding in a visible spray of water.

(b) **v. Hunyuan**, A sledgehammer is swung at a wooden fence post, causing it to break and fall to the ground.

(c) **v. Cosmos-1.0**, A javelin thrown from a high vantage point lands in a rocky outcrop below, with visible dust.

Figure 3: **Comparison of Wan2.2 with other models**. The top row shows videos generated by Wan2.2: (a) For Ray2, the jetski on the left lags behind the other jetski and then starts moving backward. (b) For Hunyuan-13B, the sledgehammer deforms after the swing, and a broken wooden board appears out of nowhere. (c) For Cosmos-7B, the javelin expels sand before it even hits the ground.

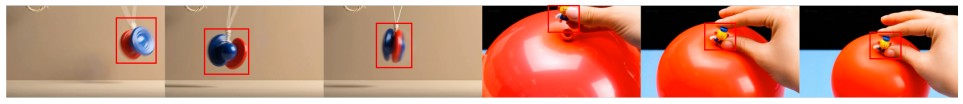

(a) A yo-yo is thrown and catches on a hanging rope; the yo-yo briefly swings back and forth before falling.

(b) A person inflates a balloon with air, then carefully pushes a small toy inside before tying it off.

Figure 4: **Illustration of Wan2.2's bad physical commonsense**. Even the best-performing model, Wan2.2, may struggle to correctly capture physical laws, leading to the generation of unnatural videos. Examples of such artifacts include: (a) A spinning yo-yo, deforming as it spins, its strings disappearing from view. (b) A balloon deflating due to an external force despite it being tied up.

ity; rather, it requires dedicated efforts to incorporate physical commonsense into video generation. Overall, our findings suggest that VIDEOPHY-2 presents a significant challenge for modern video models, with substantial room for improvement in future iterations.

**Fine-grained Analysis.** In our human annotations, we create a list of physical rules (and associated laws) that are violated in each video of the VIDEOPHY-2 dataset. We then analyze the fraction of instances in which a physical law is violated to gain fine-grained insights into model behavior. For example, if 100 physical rules are associated with the law of gravity and 25 of them are violated, the violation score would be $25\%$. We present the results of physical law violations in Figure 5. We observe that the conservation of momentum and mass are among the most frequently violated physical laws, with violation scores of $40\%$, in the videos from the VIDEOPHY-2 dataset. Conversely, we find that reflection and buoyancy are relatively mastered with violation scores less than $20\%$.[9]

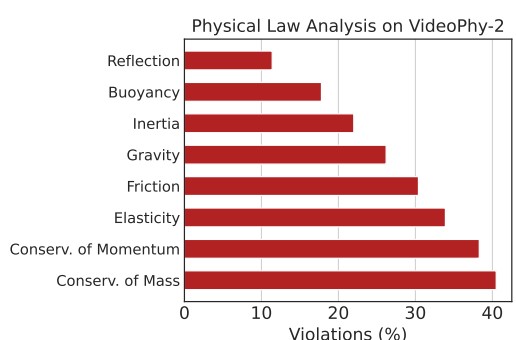

Figure 5: **Physical laws violation analysis.** We present the violation scores for diverse physical laws based on human annotations collected from various video generative models on VIDEOPHY-2.

**Qualitative Analysis.** We perform qualitative analysis to provide visual insights into the model's mode of failures. We present qualitative examples in Figure 3 to compare the best-performing model, Wan2.2-27B-A14B, with other video models. Notably, we observe violations of physical commonsense, such as jetskis moving unnaturally in reverse and the deformation of a solid sledgehammer, defying the principles of elasticity. However, even Wan lacks physical commonsense, as shown in Figure 4. In this case, we highlight that a yo-yo is shown to deform drastically as it spins, defying its material properties. Further, we cover model-specific poor physical commonsense instances along the caption and human-judged physical violations in Appendix O. For example, we show that the

---

[9]We conduct an agreement analysis across different physical laws that are derived from physical rules. We found that inter-annotator agreement scores ranged from $70\%$ to $80\%$, with lower agreement for laws like elasticity, and higher agreement (up to $80\%$) for laws such as reflection.

Table 4: **Auto-rater evaluation results.** We present the pearson's correlation ($\times 100$) between the predicted scores and ground-truth scores (1-5) on the unseen prompts and unseen video models.

| | Unseen prompts | | | Unseen video models | | |
|---|---|---|---|---|---|---|
| | Avg. | SA | PC | Avg. | SA | PC |
| VideoCon-Physics [8] | 28.5 | 32.0 | 25.0 | 26.5 | 27.0 | 26.0 |
| VideoCon [6] | 12.5 | 23.0 | 2.0 | 8.9 | 17.0 | 0.8 |
| VideoLlava [38] | 16.0 | 30.0 | 2.0 | 19.0 | 33.0 | 5.0 |
| VideoScore [23] | 13.5 | 17.0 | 10.0 | 9.0 | 5.0 | 13.0 |
| Gemini-2.0-Flash-Exp | 18.5 | 26.0 | 11.0 | 21.0 | 31.0 | 11.0 |
| Gemini-2.5-Flash-Exp | 20.5 | 31.0 | 10.0 | - | - | - |
| VIDEOPHY-2-AUTOEVAL | 42.0 | 47.0 | 37.0 | 41.0 | 45.0 | 37.0 |
| *Rel. to Best (%)* | +47.4 | +46.9 | +48.0 | +49.0 | +36.4 | +61.5 |
| *Rel. to Gemini-2.0 (%)* | +127.0 | +80.8 | +236.4 | +107.1 | +45.2 | +281.8 |

Table 5: **Auto-rater evaluation on joint score judgments.** We present the joint accuracy and F1 score between the predicted scores and ground-truth scores (0-1) for our VIDEOPHY-2-AUTOEVAL and VideoCon-Physics.

| | Unseen prompts | | | Unseen models | | |
|---|---|---|---|---|---|---|
| Method | Avg. | Acc. | F1 | Avg. | Acc. | F1 |
| VideoCon-Physics [8] | 39.1 | 75.6 | 2.6 | 39.6 | 75 | 4.2 |
| VIDEOPHY-2-AUTOEVAL | 65.1 | 79.1 | 51.1 | 62.8 | 76.3 | 49.3 |
| *Rel. to VideoCon-Physics (%)* | +66.4 | | | +49.1 | | |

Sora-generated video violates the physical rule 'The frisbee must contact the hand before any upward movement occurs' (Appendix Figure 23). We also provide several qualitative examples across diverse physical law violations across different models in Appendix P. For example, we highlight that the 'golf ball does not move after being struck by the golf club' for Ray2 (Figure 29).

## 5.2 VIDEOPHY-2-AUTOEVAL

To enable scalable judgments, we supplement the dataset with an automatic evaluator VIDEOPHY-2-AUTOEVAL. We consider two settings: (a) unseen prompts, where we assess the videos from seen video models generated using unseen (testing) prompts, and (b) unseen video models, where we assess the videos from unseen video models using unseen prompts.

We compare the correlation performance of VIDEOPHY-2-AUTOEVAL against several baselines in Table 4. In particular, VIDEOPHY-2-AUTOEVAL achieves relative gains of $47.4\%$ and $49\%$ on unseen prompts and unseen video models, respectively, compared to the best-performing baselines. Further, our auto-rater outperforms Gemini-2.0-Flash-Exp, with relative gains of $81\%$ in semantic adherence and $236\%$ in physical commonsense judgments. Further, we evaluate the accuracy and F1 performance of VIDEOPHY-2-AUTOEVAL against VideoCon-Physics for joint score judgments in Table 5. Our results show that VIDEOPHY-2-AUTOEVAL maintains a strong balance between joint accuracy and F1 scores. Finally, we assess the physical rule classification accuracy of VIDEOPHY-2-AUTOEVAL

Table 6: **Auto-rater evaluation on physical rule classification.** We present the accuracy results for VIDEOPHY-2-AUTOEVAL and other video-language models on the rule classification tasks.

| | Unseen prompts | Unseen models |
|---|---|---|
| Random | 34.5 | 31.2 |
| VideoLlava [38] | 38.1 | 38.7 |
| Gemini-2.0-Flash-Exp | 59.2 | 57.1 |
| VIDEOPHY-2-AUTOEVAL | 78.7 | 72.9 |
| *Rel. to Best (%)* | +32.9 | +27.7 |

against baselines in Table 6. Our model achieves relative gains of $32.9\%$ on unseen prompts and $27.7\%$ on unseen video models compared to Gemini-2.0-Flash-Exp. Further, we find that VIDEOPHY-2-AUTOEVAL performs much better than Gemini-2.5-Flash-Exp, consistent with our findings for Gemini-Flash-2.0. This suggests that recent improvements in state-of-the-art VLMs are not focused on enhancing semantic adherence or physical commonsense judgments for synthetic videos. Therefore, using our open and free AutoEval remains the more reliable choice for the future. Thus,

our unified auto-rater can reliably handle a variety of tasks, providing a robust tool for testing on VIDEOPHY-2.

## 6 RELATED WORK

Several works have introduced benchmarks to assess the physical plausibility of generated videos.

**VIDEOPHY** [8] was an early effort in this direction, evaluating semantic adherence and physical plausibility across 688 prompts. VIDEOPHY-2 extends this work by introducing a more extensive dataset of 3940 prompts, shifting the focus from material interactions to *real-world actions*, and providing a *5-point Likert scale* for human evaluation. Furthermore, VIDEOPHY-2 explicitly annotates physical rules in each video, allowing for a more fine-grained understanding of model behavior.

**Physics-IQ** [48] conditions models on initial frames of real videos and measures the similarity between predicted and ground-truth video continuations. While this approach offers valuable insights, it is primarily designed for video prediction rather than open-ended text-to-video generation. Additionally, extending it to longer or more complex multi-event scenarios presents challenges.

**PhyGenBench** [47] proposes a benchmark of 160 prompts and an automated evaluation framework, PhyGenEval. This work introduces structured evaluations of physical reasoning; however, its relatively small scale makes generalization difficult. Additionally, it adopts a strict one-to-one mapping between prompts and physical laws, whereas real-world physics often involves multiple interacting principles. Its evaluation relies on sequential vision-language model queries, which can introduce inconsistencies and increase computational complexity. VIDEOPHY-2 builds on these efforts by expanding dataset size, allowing for prompts that reflect multiple physical laws, and providing a more interpretable evaluation framework.

**WorldSimBench** [52] assesses video models' ability to act as *world simulators* by aligning their outputs with numerical solvers. While valuable, this approach faces challenges in bridging the *sim-to-real gap*, as physical simulations may not fully capture the complexity and variability of real-world interactions. VIDEOPHY-2 addresses this by exclusively using real-world videos to ensure that evaluations remain closely aligned with practical physical dynamics.

**Impossible Videos** [3] aims to evaluate the ability of the video generation models to create "anti or counterfactual real-world" videos (e.g., A person pours milk into a glass cup half filled with milk, but the amount of milk in the glass cup does not change at all). In contrast, our work focuses on testing the ability of the models to follow physical commonsense and semantic adherence for "real-world" actions. In this regard, the goals of these works are entirely different. Methodologically, IPV collects 260 prompts while we cover a much larger set of 4000 prompts. While IPV just evaluates "visual quality" and "semantic adherence" as Yes/No judgment from human annotators, our work performs much more fine-grained and broader evaluation. For instance, we assess the physical commonsense and semantic adherence on a scale of 1-5 which captures finer human preferences. Further, we annotate for the physical rules that are not followed in each video. In addition, we train an automatic evaluator for scalable evaluation on our dataset while IPV just tests the ability of existing VideoLLMs to solve their task instead of training their own. As a result, the practitioners will have to rely on expensive proprietary models (GPT-4o) to do automatic evaluation on IPV but our AutoEval is open and free. We present more related work in Appendix B.

## 7 CONCLUSION

We introduce VIDEOPHY-2, a benchmark for evaluating physical commonsense in videos generated by modern models. We reveal a large gap in their ability to align with prompts and generate videos that follow physical commonsense. Further, we provide physical law violations and an auto-rater for scalable evaluation. Overall, this dataset advances our understanding of the current state of the video generative models as general-purpose world simulators.

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
