## A    LLM USAGE

In this work, we used large language models (LLMs), such as the ChatGPT API, as writing assistants to proofread our self-written content and correct any grammatical issues. LLMs were not used for research ideation or for introducing new explanations to our experimental findings.

## B    MORE RELATED WORK

The rapid advancement of video generation models necessitates robust benchmarks and evaluation methodologies, particularly for assessing *physical commonsense* understanding. This is a crucial step towards realizing the vision of video generation models as 'world simulators' [13]. Our work, VIDEOPHY-2, extends prior work by addressing key limitations in existing benchmarks and evaluation methods. We structure our discussion of related work around the following key areas:

### B.1    VIDEO GENERATION MODELS

Recent progress in text-to-video (T2V) generation has been driven primarily by two architectural paradigms: diffusion models [24, 25, 12, 62, 61, 29] and autoregressive models [45, 33, 26, 59]. Diffusion models, such as Stable Video Diffusion (SVD) [11] and Sora [42], involve a multi-stage training process and operate in a latent space. Autoregressive models, including VideoPoet [33] and CogVideo [26], predict future frames based on past frames. While these models demonstrate impressive visual fidelity, their ability to capture underlying physical principles remains an open question. This highlights the need for rigorous benchmarks like VIDEOPHY-2. Additionally, recent works, such as Genie [14], explore interactive video generation, further expanding the potential applications of these models beyond static content creation.

### B.2    GENERAL VIDEO GENERATION BENCHMARKS

Several benchmarks focus on evaluating general aspects of video quality but do not specifically assess physical reasoning. **VBench** [28] introduces a hierarchical evaluation approach, covering motion smoothness, background consistency, and overall visual fidelity. **EvalCrafter** [41] proposes 17 objective metrics focused on different aspects of video quality. While these benchmarks contribute to understanding model performance, they do not isolate or systematically evaluate physical commonsense in generated videos.

### B.3    AUTOMATIC EVALUATION METHODS

Traditional video quality metrics, such as Fréchet Video Distance (FVD) [57], were not designed to assess physical plausibility. More recent approaches incorporate vision-language models (VLMs) for automatic evaluation [23, 8, 39]. VIDEOPHY introduced a fine-tuned VLM-based evaluator, but existing VLMs still face challenges in reliably assessing physical commonsense. VIDEOPHY-2 proposes VIDEOPHY-2-AutoEval, an enhanced automatic evaluator trained on a larger and more diverse dataset with explicit physical rule annotations. This leads to improved alignment with human assessments while maintaining interpretability.

### B.4    PHYSICAL REASONING IN AI

The study of physical commonsense in AI builds upon insights from both artificial intelligence and cognitive science. Research on intuitive physics has explored how humans reason about object interactions and causal physical events [46, 9, 20]. In AI, approaches such as PIQA [10] assess physical reasoning in language, while others focus on synthetic environments for learning physical interactions [4, 19, 15, 1]. VIDEOPHY-2 contributes to this area by providing a large-scale, real-world benchmark that bridges the gap between theoretical physical reasoning and practical video generation.

In summary, VIDEOPHY-2 advances the evaluation of physical commonsense in video generation by introducing a large-scale, action-centric benchmark, a fine-grained evaluation framework with explicit rule annotations, and a robust, human-aligned automatic evaluator. These contributions address key

Table 7: **Data statistics.** We present the number of instances for diverse features (e.g., captions,) in the VIDEOPHY-2.

| Feature | Number |
|---|---|
| Captions | 3940 |
| Unique actions | 197 |
| Generated videos | 8000 |
| Human annotations | 110K |
| Avg. words in original caption | 16 |
| Avg. words in upsampled caption | 138 |
| *Category* | *# Actions* |
| Sports and Physical Activities | 143 |
| Object Interactions | 54 |
| Hard subset | 60 |

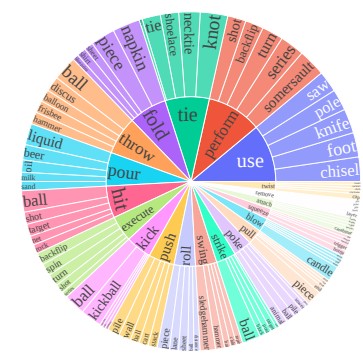

Figure 6: **Diversity of VIDEOPHY-2 prompts.** Top-20 frequently occurring verbs (inner) and their top-5 direct nouns (outer).

limitations of prior work and facilitate a more comprehensive assessment of video models' ability to simulate the physical world.

## C    LIST OF ACTIONS

Our benchmark includes a broad set of actions spanning simple motions (e.g., *walking through snow, dribbling basketball*), object interactions (e.g., *peeling fruit, using a paint roller*), and complex physical activities (e.g., *pole vault, tightrope walking*). These actions ensure models are tested across diverse movement dynamics, real-world interactions, and varying levels of physical complexity.

Table 9 presents the full set of 197 actions.

## D    LIST OF HARD ACTIONS

Certain actions pose significant challenges for generative models due to rapid motion (e.g., *throwing discus, gymnastics tumbling*), intricate interactions (e.g., *popping balloons, pouring until overflow*), and structural deformations (e.g., *ripping paper, bending until breaking*).Actions were deemed hard using CogVideoX-5B as a baseline.

Table 10 presents a subset of 60 hard actions, ensuring our benchmark tests model capabilities beyond simple motions, including stability, object manipulation, and real-world physics.

## E    DIVERSITY OF PROMPTS FOR DIVERSE ACTIONS

In this work, our objective was to curate a set of actions from which we could generate a wide range of diverse prompts that, while focusing on motion, still encompass various real-world settings. By ensuring diversity in prompt selection within each action, our dataset becomes more robust and capable of testing generative models across a broad spectrum of physical interactions.

Figure 7 illustrates a subset of actions used in our study, along with the verb-noun pairs appearing in their respective generated prompts. The diversity of nouns and verbs in these prompts ensures that our dataset covers a wide range of contextual variations. For instance, the action *Rowing* can be associated with nouns such as *boat*, *paddle*, or *water*, leading to prompts that span different environments and interactions. Similarly, *Knitting* may involve *yarn*, *needles*, or *fabric*, allowing for a richer set of generated scenarios that helps to better evaluate video-generation models.

## F    LLM-GENERATED CAPTIONS PROMPT

In this section, we discuss the prompt we gave Gemini-2.0-Flash-Exp to generate a list of diverse prompts for each action, displayed in Table  10.

## G  VIDEO-SPECIFIC PHYSICAL RULE GENERATION PROMPT

In this section we discuss the prompt we gave Gemini-2.0-Flash-Exp to generate a list of 3 rules for each prompt, displayed in Table 11.

## H  UPSAMPLED CAPTION EXAMPLES

To enhance video generation quality, we upsample captions by upsampling them to make them more specific with finer details. This process helps models generate better videos by providing additional cues about motion, environment, and object interactions.

Table 14 presents examples where simple captions (e.g., *"A person uses nunchucks to break a stack of wooden blocks"*) are expanded into more vivid descriptions (e.g., including details about lighting, camera angles, and material properties). These enriched captions guide models toward generating more coherent and contextually accurate videos.

## I  TRAINING DETAILS FOR VIDEOPHY-2-AUTOEVAL

We finetune VideoCon-Physics [8] (7B) that acts a strong base model for semantic adherence and physical commonsense evaluation. Specifically, we use low rank adaptation [27] applied to all the transformer blocks including QKVO, gate, up, and down weight matrices. In our experiments, we set $r, \alpha = 32$ and dropout=0.05. We finetune the base model for 3 epochs and pick the best checkpoint using the performance on the validation set. Further, we use Adam optimizer [30] with a linear warmup steps of 50 steps followed by linear decay. We perform a hyperparameter search over several peak learning rates $\{5e-5, 1e-4, 5e-4, 1e-3\}$ and found $5e-4$ worked the best. We use 4 Nvidia A6000 GPUs with a global batch size of 64.

## J  MULTIMODAL PROMPTS FOR AUTOMATIC EVALUATION

We prompt our model to generate a text response conditioned on the multimodal template $\mathcal{T}_t(x)$ for semantic adherence, physical commonsense, and rule tasks. Formally,

$$\mathcal{T}_t(x) = \begin{cases} \mathcal{T}_{SA}(V, C), & t = SA \\ \mathcal{T}_{PC}(V), & t = PC \\ \mathcal{T}_R(V, R), & t = RS \end{cases} \tag{1}$$

where $t$ is either semantic adherence to the caption, physical commonsense, or the rule score, $C$ is the conditioning caption, $V$ is the generated video for the caption $C$, and $R$ is the rule candidate. We provide the multimodal templates ($\mathcal{T}_{SA}(V, C)$, $\mathcal{T}_{PC}(V)$, and $\mathcal{T}_R(V, R)$). We compute the score from the model using simple autoregressive generation.

We present the prompts used for the this multimodal evaluation for semantic adherence evaluation in Figure 14, physical commonsense alignment in Figure 15, and rule scoring in Figure 16.

## K  BASELINES JUDGMENTS

### K.1  VIDEOPHYSICS AND VIDEOCON

For our baseline comparisons, we obtained raw scores from VideoPhysics[8] and VideoCon[6] frameworks, which originally produced scores in the range of 0-1. To maintain consistency with our 1-5 rating scale, we normalized these scores using linear scaling. Specifically, we applied the transformation $score_{normalized} = score_{raw} \times 4 + 1$, followed by rounding to the nearest integer. This transformation maps the minimum possible score (0) to 1 and the maximum possible score (1) to 5, preserving the relative performance differences between models while making the scores directly comparable to our human evaluation ratings.

Table 8: **Inference settings for different video generation models.**

| Models | Caption Type | FPS/Video Length (s) | Resolution | Frames | Scale Guidance | Steps Sampling | Scheduler Noise | Precision |
|---|---|---|---|---|---|---|---|---|
| Wan2.1-14B | Upsampled | 16 (4s) | 832×480 | 61 | 5 | 50 | FlowUniPCMultistepScheduler | bf16 |
| Wan2.2-T2V-27B-A14B | Upsampled | 16 (4s) | 832×480 | 61 | 5 | 50 | FlowUniPCMultistepScheduler | bf16 |
| CogVideoX-5B | Upsampled | 8 (6s) | 480×720 | 49 | 6 | 50 | CogVideoXDPMScheduler | bf16 |
| Cosmos-Diffusion-7B | Upsampled | 24 (5s) | 576×576 | 120 | 7 | 60 | - | bf16 |
| HunyuanVideo-13B | Original | 15 (4s) | 320×512 | 61 | 6 | 50 | FlowMatchEulerDiscreteScheduler | fp16 |
| VideoCrafter2-1.4B | Original | 10 (3s) | 320×512 | 32 | 12 | 50 | DDIM | fp32 |
| StableVideoDiffusion | Original | 4 (4s) | 576x1024 | 14 | - | 25 | EDM | fp16 |

## K.2 VIDEOSCORE REGRESSION

For our implementation of VideoScore [23], we selected specific component metrics that align with our evaluation objectives. For SA, we utilized the Text-to-Video Alignment score, as it effectively measures the correspondence between video content and textual descriptions. For PC, we employed the Factual Consistency Score, which assesses the physical plausibility of events depicted in the videos.

## K.3 VIDEOLLAVA

For VideoLlava [38], we used the same prompts shown in Appendix J. The model produced scores directly on our 1-5 scale following the evaluation criteria provided in the prompts.

## K.4 GEMINI

For Gemini-2.0-Flash-Exp, we leveraged its larger context window to provide more detailed and structured evaluation prompts, allowing us to specify evaluation criteria with greater granularity, and adding few-shot examples. This enabled more comprehensive assessments for semantic adherence (Figure 12) and physical commonsense (Figure 13).

## L HUMAN ANNOTATION INTERFACE

Figure 17 and Figure 18 showcase our human annotation interface. The interface is designed to facilitate both semantic adherence and physical commonsense evaluation. We combine rule scoring and physical commonsense assessment into a single task, allowing annotators to provide additional rules they believe are violated.

## M MODEL INFERENCE DETAILS

Table 8 summarizes the inference settings for the video generation models used in our study. The table highlights key parameters such as resolution, frame rate, guidance scale, sampling steps, and precision. In our experiments, all models except those with a token limit of 77 (due to CLIP [53] embeddings) used upsampled captions. Models like Wan2.1/2.2 and CogVideoX-5B particularly benefit from these richer descriptions, enhancing the quality of their generated videos. We evaluated the models using original and upsampled captions. Different models use different schedulers, and use different default precision levels (bf16, fp16, fp32). For closed models, such as Ray2 and Sora, we used fewer captions, which influences their evaluation outcomes.

## N DISTRIBUTION OF SEMANTIC ADHERENCE AND PHYSICAL COMMONSENSE SCORES

We present the distribution of semantic adherence (SA) and physical commonsense (PC) scores in Figure 8 and 9.

mopping floor, blowing out candles, throwing water balloon, passing american football, billiards, lifting a surface with something on it until it starts sliding down, using a sledge hammer, swing, hitting baseball, hammerthrow, playing tennis, longjump, sewing, roller skating, wading through water, riding mechanical bull, pole vault, blowdryhair, tying necktie, paragliding, something falling like a rock, playing ice hockey, sailing, gymnastics tumbling, pushing something so that it slightly moves, folding clothes, poking something so lightly that it doesn't or almost doesn't move, javelinthrow, surfing, snatch weight lifting, chiseling stone, spinning something that quickly stops spinning, poking a stack of something without the stack collapsing, carving ice, throwing something, twisting (wringing) something wet until water comes out, putting something onto something else that cannot support it so it falls down, wiping something off of something, playing field hockey, juggling balls, wading through mud, shooting basketball, welding, hoverboarding, javelin throw, catching or throwing softball, hammering, knitting, putting something that can't roll onto a slanted surface so it slides down, playing polo, pouring something onto something, pulling two ends of something so that it gets stretched, using circular saw, flint knapping, backflip (human), long jump, uncovering something, pouring something into something until it overflows, dribbling basketball, poking a hole into some substance, bulldozing, peeling fruit, parallelbars, playing darts, spinning something so it continues spinning, luge, pushing something so it spins, curling (sport), riding unicycle, throwing discus, folding paper, ripping paper, trapezing, playing pinball, burying something in something, throwing axe, wrapping present, yarn spinning, tying shoe laces, flying kite, tightrope walking, using a paint roller, using a wrench, sharpening pencil, pizzatossing, catching or throwing baseball, catching or throwing frisbee, playing kickball, golf, nunchucks, pouring something out of something, opening bottle (not wine), unfolding something, stuffing something into something, canoeing or kayaking, rolling something on a flat surface, playing ping pong, punching bag, picking fruit, poking something so that it spins around, balancebeam, parasailing, jumprope, bungee jumping, drop kicking, hammer throw, using segway, biking through snow, swimming, making snowman, rowing, attaching something onto something, something colliding with something and both come to a halt, blasting sand, throwdiscus, tying knot (not on a tie), digging something out of something, trimming shrubs, inflating balloons, bouncing on trampoline, spinning poi, shot put, pushing something onto something, something falling like a feather or paper, letting something roll down a slanted surface, cutting, chopping wood, breaststroke, hurdling, ice climbing, popping balloons, throwing knife, bending something so that it deforms, playing cricket, shaping bread dough, bobsledding, smashing, blowing bubble gum, something colliding with something and both are being deflected, breaking boards, somersaulting, skateboarding, squeezing something, ropeclimbing, bending something until it breaks, yoyo, bouncing on bouncy castle, hula hooping, letting something roll along a flat surface, pushing something off of something, lifting a surface with something on it but not enough for it to slide down, throwing snowballs, shoveling snow, rolling pastry, tossing coin, threading needle, skijet, clay pottery making, pommelhorse, playing squash or racquetball, pushing something so that it falls off the table, bodysurfing, twisting something, poking a hole into something soft, tearing something into two pieces, walking through snow, putting something that cannot actually stand upright upright on the table so it falls on its side, poking something so that it falls over, poking a stack of something so the stack collapses, pushing something so that it almost falls off but doesn't, kicking soccer ball, tying bow tie, wood burning (art), putting something that can't roll onto a slanted surface so it stays where it is, extinguishing fire, ice skating, playing badminton, archery, folding napkins, soccerjuggling, blowing leaves, bowling, mountain climber (exercise), jetskiing, pouring something into something, pouring beer, pulling two ends of something so that it separates into two pieces, volleyballspiking, poking something so it slightly moves, smoking, tearing something just a little bit, rock climbing, letting something roll up a slanted surface so it rolls back down, blowdrying hair, folding something, riding scooter, playing volleyball

Table 9: Actions List: A comprehensive collection of all physical actions that were used test video models' ability to produce videos that align with physical commonsense

```
blowing out candles, playing squash or racquetball, passing
american football, billiards, backflip (human), tearing something
into two pieces, pouring something into something until it
overflows, lifting a surface with something on it until it starts
sliding down, using a sledge hammer, swing, hitting baseball,
peeling fruit, parallelbars, putting something that cannot
actually stand upright upright on the table so it falls on its
side, playing darts, poking something so that it falls over,
chopping wood, throwing discus, poking a stack of something so
the stack collapses, pushing something so that it almost falls off
but doesn't, kicking soccer ball, pole vault, popping balloons,
ripping paper, blowdryhair, throwing knife, playing ice hockey,
throwing axe, bending something so that it deforms, yarn spinning,
playing cricket, tightrope walking, gymnastics tumbling, playing
badminton, archery, pizzatossing, catching or throwing baseball,
catching or throwing frisbee, chiseling stone, spinning something
that quickly stops spinning, folding napkins, golf, throwing
something, nunchucks, opening bottle (not wine), bending something
until it breaks, wiping something off of something, playing field
hockey, balancebeam, pushing something off of something, throwing
snowballs, pouring beer, bungee jumping, drop kicking, pulling two
ends of something so that it separates into two pieces, catching
or throwing softball, rowing, hammering, letting something roll up
a slanted surface so it rolls back down, playing polo
```

Table 10: Hard Actions List: A comprehensive collection of challenging physical actions in the "Hard" category used to test video models' ability to produce videos that align with physical commonsense

Table 11: **Semantic adherence evaluation results across models.**

| Model | SA | | | |
|---|---|---|---|---|
| | All | Hard | Physical Activities | Object Interactions |
| Wan2.2-27B-A14B | **3.8** | **3.7** | **3.8** | **4.0** |
| Wan2.1-14B | 3.3 | 3.1 | 3.3 | 3.5 |
| CogVideoX-5B | 3.0 | 2.4 | 3.0 | 3.0 |
| Cosmos-Diff-7B | 3.2 | 2.8 | 3.2 | 3.3 |
| Hunyuan-13B | 2.9 | 2.7 | 2.9 | 2.9 |
| VideoCrafter-2 | 2.6 | 2.4 | 2.6 | 2.6 |
| SVD-I2V | 2.1 | 2.0 | 1.9 | 2.4 |
| Ray2 | 3.1 | 2.8 | 3.1 | 3.1 |
| Sora | 2.8 | 2.3 | 2.7 | 3.0 |

## O   POOR PHYSICAL COMMONSENSE QUALITATIVE EXAMPLES BY MODEL

We present more examples from each generative model where one or more physical laws are violated in Figure 19 - Figure 28.

## P   POOR PHYSICAL COMMONSENSE QUALITATIVE EXAMPLES BY LAW

We present a few qualitative examples highlighting instances where specific physical laws are violated in Figure 29 - Figure 36.

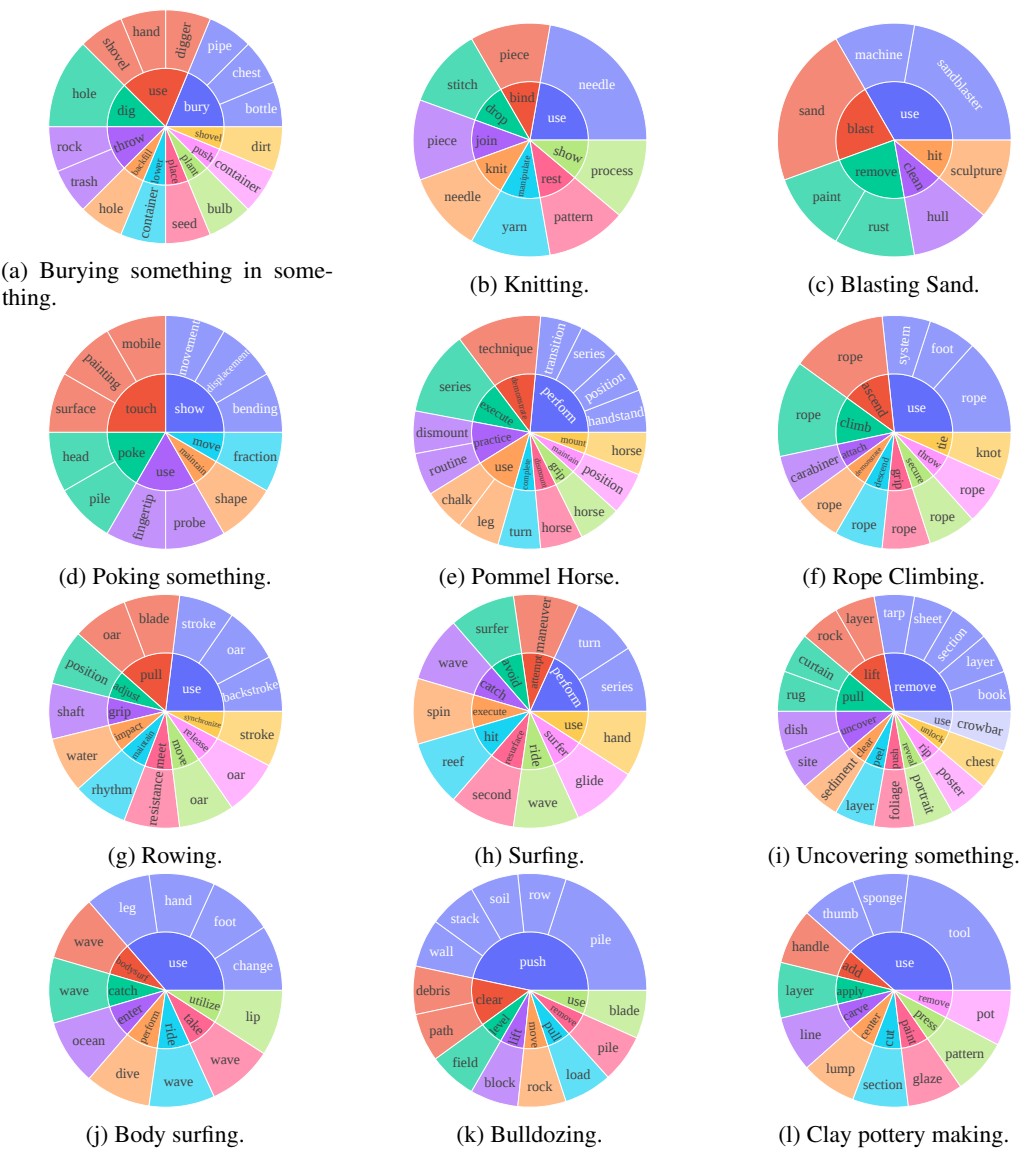

Figure 7: A subset of actions, and the verb-noun pairs in their respective generated prompts.

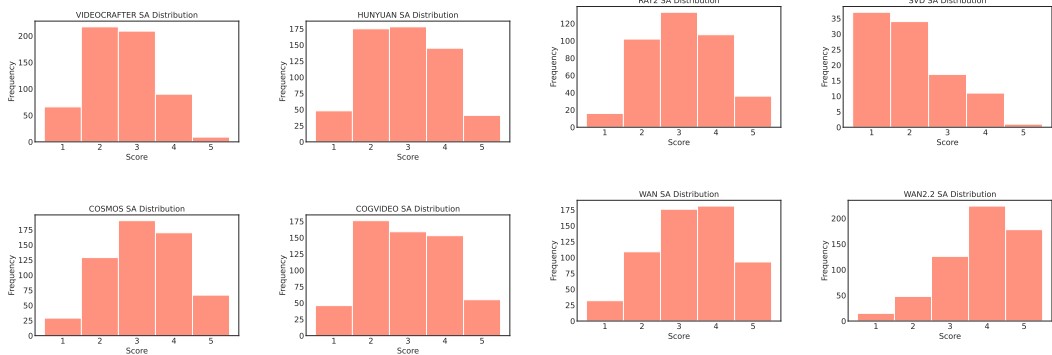

Figure 8: Distribution of semantic adherence scores for various models in VIDEOPHY-2.

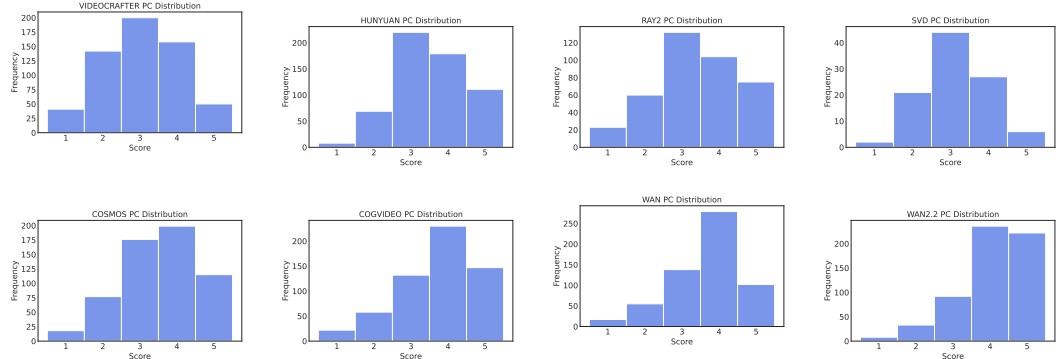

Figure 9: Distribution of physical commonsense scores for various models in VIDEOPHY-2.

Table 12: **Physical commonsense evaluation results across models.**

| Model | PC | | | |
|---|---|---|---|---|
| | All | Hard | Physical Activities | Object Interactions |
| Wan2.2-27B-A14B | **4.1** | **4.0** | **4.1** | **4.0** |
| Wan2.1-14B | 3.7 | 3.4 | 3.6 | 3.7 |
| CogVideoX-5B | 3.7 | 3.3 | 3.6 | 3.9 |
| Cosmos-Diff-7B | 3.5 | 3.2 | 3.5 | 3.5 |
| Hunyuan-13B | 3.5 | 3.3 | 3.5 | 3.7 |
| VideoCrafter-2 | 3.1 | 2.9 | 3.0 | 3.3 |
| SVD-I2V | 3.1 | 3.0 | 3.2 | 3.1 |
| Ray2 | 3.4 | 3.2 | 3.3 | 3.5 |
| Sora | 3.6 | 3.5 | 3.6 | 3.6 |

Task: Generate 20 realistic, detailed, and diverse video prompts based on a given action. The prompts must focus solely on clear, observable physical actions and interactions between characters, objects, and their environments that can be easily represented visually in a video.
**Requirements:**
1. **Physical Actions with Direct Outcomes:** Each prompt must clearly describe observable physical actions involving objects (e.g., bows, arrows, targets, or archers) and their direct, visible outcomes (e.g., an arrow striking a target, string tension when pulled).
2. **Exclude Non-Visual Details:** Avoid including sensory or inferred details like sounds, smells, emotions, or mental states (e.g., "focus," "determination," or "excitement").
3. **Avoid Subtle Movements:** Do not include descriptions of small-scale or subtle motions that are hard to detect in standard video playback (e.g., trembling hands, tiny vibrations, or imperceptible shifts).
4. **Concrete, Not Abstract:** Steer clear of poetic, artistic, or abstract descriptions (e.g., "the shimmering flight of an arrow" or "a poised stance") and instead focus on tangible, visible actions.
5. **Diverse Scenarios:** Use a variety of settings, character types, objects, and equipment to make the prompts diverse and adaptable to different video generation contexts.
6. **Specific Visual Actions:** Center prompts on observable, specific actions such as pulling a bowstring, releasing an arrow, an arrow's flight path, or its interaction with targets. Highlight the physical interactions between objects (e.g., wood, metal, or other materials) and environments.
**Examples:**
**Good Prompts for the Action "Archery"**
"An archer draws the bowstring back to full tension, then releases the arrow, which flies straight and strikes a bullseye on a paper target."
"A compound bow is fired, and the arrow pierces through a stack of hay bales, stopping halfway through."
"An archer adjusts their stance, takes aim, and releases an arrow, which embeds itself into a foam target with visible force."
"A crossbow is loaded with a bolt, cocked, and fired, hitting a glass bottle, which shatters on impact."
"An arrow is shot toward a wooden target, splinters flying as it embeds deep into the surface."
**Bad Prompts for the Action "Archery"**
"The archer feels confident as they aim their arrow at the target." ## (Describes inferred mental state.)
"The arrow flies silently and gracefully toward the target." ## (Includes non-visual elements and artistic descriptions.)
"The string vibrates slightly after the arrow is released." ## (Focuses on subtle, hard-to-detect motion.)
"The archer holds their bow with a poised and elegant stance." ## (Focuses on posture instead of action.)
Now, Generate up to 20 prompts relevant to the given action: "" while adhering to the above criteria. Use various objects, environments, and physical interactions to ensure diversity and realism. Output in a python-parsable list of strings, stored in the variable "'prompts'".

Figure 10: LLM Prompt for Generating Realistic Video Captions.

**Task Description:** Generate simple, clear behaviors/rules for describing visible, real-world physical interactions in a given video scene. These behaviors will be used to determine whether a video aligns with realistic physical interactions.

**Key Requirements:**

1. **Observation-Centric:** Focus strictly on what is visually observable in the video (e.g., motion, deformation, changes in shape or position). Avoid abstract concepts or invisible factors (e.g., forces, emotions, intentions).

2. **Action-Oriented:** The behaviors should directly describe interactions between materials or objects, not actions or intentions of individuals. The focus should be on the materials' behavior (e.g., how the gauze behaves when wrapped, how it stretches, or conforms), not the motions of the hands or other actors.

3. **Simple and Testable:** The behaviors must be concise, clear, and directly testable from the video. Avoid jargon or unnecessary technical details.

4. **Associated Physical Laws:** Each rule should include the physical law(s) it exemplifies. Use applicable laws from the following list:

> *"Gravity", "Buoyancy", "Elasticity", "Friction", "Conservation of Mass",*
> *"Reflection", "Refraction", "Interference and Diffraction", "Tyndall Effect"*
> *"Sublimation", "Melting", "Boiling", "Liquefaction"*
> *"Hardness", "Solubility", "Dehydration Properties", "Flame Reaction"*

Other laws are acceptable (e.g., Archimedes' Principle), but they must be well-known physical laws. Rules should only be related to visible, observable physical properties of materials, such as shape, deformation, or material properties, and not vague statements that are not strictly physical phenomena.

5. **Guaranteed:** The behaviors must occur in the video as specified from the prompt description. Events that are not guaranteed to occur from the prompt description should not appear in the rules list. Do not add information beyond what is present in the caption.

**Example Prompt:** *A basketball bounces up and down*

**Good Examples of Behaviors:**
- The ball is faster at the bottom of the bounce. (Gravity)
- The ball moves up after bouncing off the floor. (Elasticity)

**Bad Examples of Behaviors:**
- The gravity acts on the ball. (## Not specific enough to test.)
- The ball deforms when it collides with the floor. (## Difficult to test.)
- The ball is caught if it is being dribbled. (## Not guaranteed from the prompt.)
- The ball stops bouncing after a while. (## Not guaranteed; the prompt does not specify video duration.)

**Example Prompt:** *A nurse applies a bandage around a patient's arm.*

**Good Examples of Behaviors:**
- The bandage is a flexible and solid material. (Elasticity)
- The bandage roll becomes smaller as it is unrolled. (Conservation of Mass)

**Bad Examples of Behaviors:**
- The bandage stops the bleeding. (## No bleeding mentioned in the prompt.)
- The bandage secures the wound tightly. (## Tightness is subjective and not visually testable.)
- The bandage is elastic and stretches when pulled. (## Elasticity applies, but not directly related to the video's action.)

Now, generate a suitable list of 3 behaviors for this prompt:

**A person shoots an explosive arrow at a metal target.**

Output in a Python-comprehensible format with:

`[("rule1", ["Melting","Gravity"]), ("rule2", ["Conservation of Momentum"]), ..., etc.]`

Stored in the variable: `behaviors`.

Figure 11: LLM Prompt for Generating Physical Rules.

Table 13: **Sample prompts in the VIDEOPHY-2.** We present the action and prompt, along with their category—either focusing on physical activities or object interactions. Additionally, we highlight the potential physical principle associated with each prompt.

| Action | Prompt | Physical Principle | Category |
|---|---|---|---|
| Canoeing | A person uses a kayak paddle to push their kayak up the bank of a river. | Buoyancy | Physical Activity |
| Riding Unicycle | A person on a unicycle stops abruptly, putting a foot down to regain balance. | Inertia | Physical Activity |
| Pushing something so it spins | A chef pushes a pizza spinning on a tray with their hand. | Conversation of Momentum | Object Interaction |

| Original Caption | Upsampled Caption |
|---|---|
| A person uses nunchucks to break a stack of wooden blocks. | In a dynamic display of martial arts prowess, a skilled practitioner wields a pair of nunchucks, their hands clad in black gloves that enhance grip and safety. The scene unfolds in a dimly lit, industrial setting, where a stack of wooden blocks, meticulously arranged in a pyramid, awaits the test of strength. The camera captures the moment with a static focus, highlighting the nunchucks' fluid motion as they swing through the air, their polished wooden surfaces glinting under the soft, ambient light. |
| A player throws a softball sidearm, and the ball spins as it travels through the air. | In a sun-drenched outdoor sports arena, a dynamic softball game unfolds, captured with cinematic precision. The camera, positioned at a low angle, focuses on a player in a sleek black jersey, their arm poised in a powerful sidearm throwing motion. The softball, a vibrant white against the lush green field, spins rapidly as it leaves their hand, tracing a graceful arc through the air. |
| A small rock tumbles down a steep, rocky hillside. | In a breathtaking display of nature's raw power, a small, dark rock careens down a steep, rugged hillside, its descent punctuated by a cascade of displaced soil and smaller stones. The camera, positioned at a static angle, captures the dynamic interplay of gravity and friction as the rock bounces and rolls, kicking up a cloud of dust that dances in the sunlight. |
| An athlete throws a hammer, with the hammer head clearly visible in rotation. | In a breathtaking outdoor setting, a skilled athlete stands poised, clad in a sleek black and white athletic uniform, their focus unwavering as they prepare to hurl a gleaming hammer. The hammer rotates gracefully, its head spinning in a mesmerizing blur against the backdrop of a clear blue sky. |
| A person on a unicycle stops abruptly, putting a foot down to regain balance. | In a serene outdoor setting, a solitary figure clad in a vibrant red jacket and black pants gracefully navigates a narrow, winding path on a sleek black unicycle. The camera captures the scene with a static shot, allowing the viewer to fully appreciate the rider's skill and balance as they glide effortlessly forward. |

Table 14: Example of upsampled captions given to video generation models

---

**Task Description:** Evaluate the alignment between a video and its accompanying caption: *"A badminton shuttlecock is served underhand, traveling across the net and landing within the service box."*
**Evaluation Criteria:**
1. **Entities and Objects:** Do the objects, entities, and subjects mentioned in the caption appear in the video?
2. **Actions and Events:** Are the actions, events, or interactions described in the caption clearly depicted in the video?
3. **Temporal Consistency:** If the caption describes a sequence or progression of events, does the video follow the same temporal order?
4. **Scene and Context:** Does the overall scene (e.g., background, setting) match the description in the caption?
**Instructions for Scoring:**
- **1:** No alignment. The video does not match the caption at all (e.g., different objects, events, or scene).
- **2:** Poor alignment. Only a few elements of the caption are depicted, but key objects or events are missing or incorrect.
- **3:** Moderate alignment. The video matches the caption partially, but there are inconsistencies or omissions.
- **4:** Good alignment. Most elements of the caption are depicted correctly in the video, with minor issues.
- **5:** Perfect alignment. The video fully adheres to the caption with no inconsistencies.
**Example Prompt:** *"A badminton shuttlecock is served underhand, traveling across the net and landing within the service box."*
**Example Responses:**
**Score:** 3 **Explanation:** The shuttlecock is present, and an underhand serve is performed. However, the landing position is unclear, and the trajectory is partially obstructed.

Figure 12: Gemini Prompt for Evaluating Semantic Adherence.

**Task Description:** Evaluate whether the video follows physical commonsense. This judgment is based solely on the video itself and does not depend on the caption.

**Evaluation Criteria:**

1. **Object Behavior:** Do objects behave according to their expected physical properties (e.g., rigid objects do not deform unnaturally, fluids flow naturally)?

2. **Motion and Forces:** Are motions and forces depicted in the video consistent with real-world physics (e.g., gravity, inertia, conservation of momentum)?

3. **Interactions:** Do objects interact with each other and their environment in a plausible manner (e.g., no unnatural penetration, appropriate reactions on impact)?

4. **Consistency Over Time:** Does the video maintain consistency across frames without abrupt, unexplainable changes in object behavior or motion?

**Instructions for Scoring:**

- **1:** No adherence to physical commonsense. The video contains numerous violations of fundamental physical laws.
- **2:** Poor adherence. Some elements follow physics, but major violations are present.
- **3:** Moderate adherence. The video follows physics for the most part but contains noticeable inconsistencies.
- **4:** Good adherence. Most elements in the video follow physical laws, with only minor issues.
- **5:** Perfect adherence. The video demonstrates a strong understanding of physical commonsense with no violations.

**Example Responses:**

**Score:** 2 **Explanation:** The ball's motion is inconsistent with gravity; it hovers momentarily before falling. Additionally, object interactions lack expected momentum transfer, suggesting physics inconsistencies.

Figure 13: Gemini Prompt for Evaluating Physical Commonsense.

**Semantic Adherence**:

**Given: V** (Video), **T** (Caption)

**Instruction (I):** *[V] Does this video match the description: "[T]"?*
*Please rate the video on a scale from 1 to 5, where 5 indicates a perfect match and 1 indicates no relevance.*

**Response (R):** *1, 2, 3, 4, or 5*

Figure 14: Template for assessing semantic adherence using a multi-modal VLM.

**Physical Commonsense**:

**Given: V** (Video)

**Instruction (I):** *[V] Does this video adhere to physical laws?*
*Rate the video on a scale from 1 to 5, where 5 means full compliance and 1 means significant violations.*

**Response (R):** *1, 2, 3, 4, or 5*

Figure 15: Template for assessing physical commonsense in video-based evaluation.

**Rule Validation**:

**Given: V** (Video), **R** (Physical Rule)

**Instruction (I):** *[V] Does the video follow the physical rule: "[R]"?*
*Choose 0 if not, 1 if valid, or 2 if indeterminate.*

**Response (R):** *0, 1, or 2*

Figure 16: Template for validating specific physical rules in a video.

Answer the following questions based on the description and the AI-generated video.

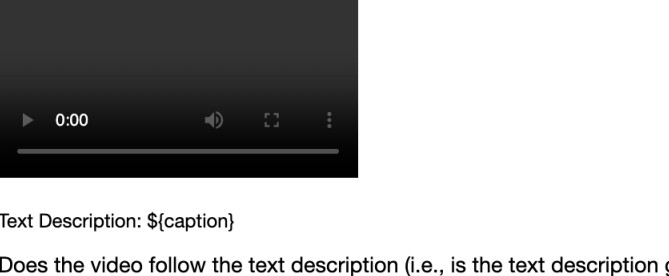

Text Description: ${caption}

Does the video follow the text description (i.e., is the text description grounded in the video)?

○ Very Unlikely
○ Unlikely
○ Neutral
○ Likely
○ Very Likely

**Submit**

Figure 17: The screenshot of the human annotation interface for semantic adherence task.

Answer the following questions about the AI-generated video.

Is the physical rule "**${rule_1}**" valid for the given video?

○ Yes
○ No
○ Cannot be determined from the video

Is the physical rule "**${rule_2}**" valid for the given video?

○ Yes
○ No
○ Cannot be determined from the video

Is the physical rule "**${rule_3}**" valid for the given video?

○ Yes
○ No
○ Cannot be determined from the video

Does the video follow physical commonsense (e.g., does not violate commonsense and laws of physics)?

○ Very Unlikely
○ Unlikely
○ Neutral
○ Likely
○ Very Likely

If a physical rule is **violated** in the video but is not included in the list of rules, please describe the missed rule briefly in 1-2 lines.

Figure 18: The screenshot of the human annotation interface for physical rule and commonsense judgment tasks.

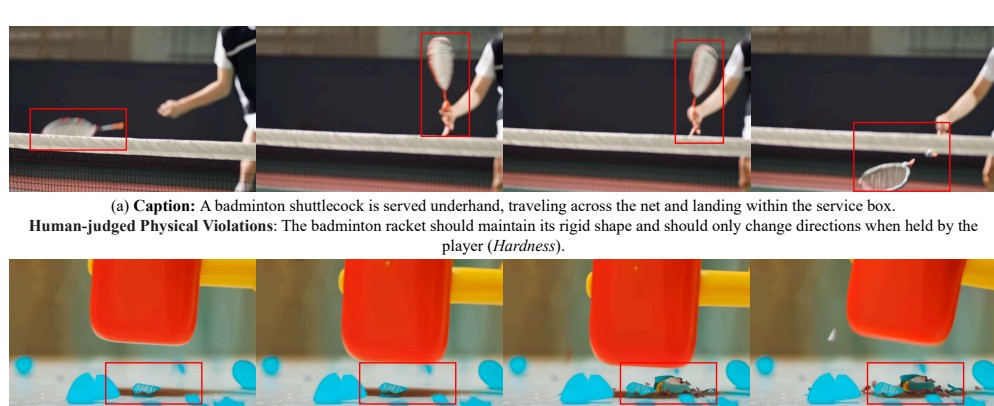

(a) **Caption:** A badminton shuttlecock is served underhand, traveling across the net and landing within the service box.
**Human-judged Physical Violations**: The badminton racket should maintain its rigid shape and should only change directions when held by the player (*Hardness*).

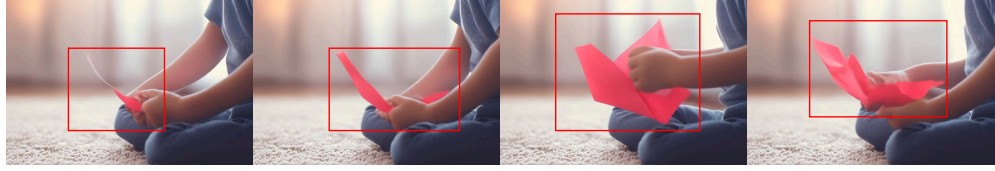

(b) **Caption:** A child's toy hammer smashes a small plastic egg, breaking it open.
**Human-judged Physical Violations**: The mass and amount of shards of egg should stay constant, no mass should appear out of nowhere (*Conservation of Mass*).

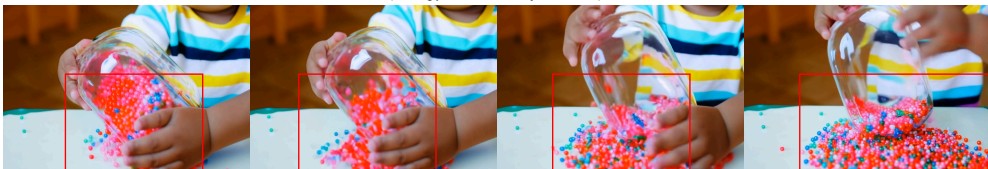

(c) **Caption:** A child folds a piece of origami paper into a simple crane, with visible creases appearing.
**Human-judged Physical Violations**: Creases should not spontaneously form in the paper without an external force or proper folding sequence (*Entropy, Material Deformation*).

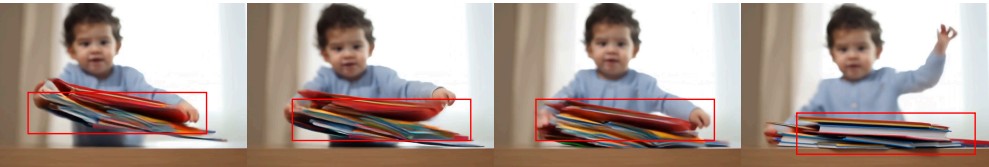

(d) **Caption:** A child pours colorful beads from a plastic container into a glass jar until they overflow, scattering on the floor.
**Human-judged Physical Violations**: The container should not leak beads with no openings (*Gravity, Impermeability of Solids*).

(e) **Caption:** A child pushes a stack of books off a desk; the books fall in a chaotic pile.
**Human-judged Physical Violations**: The amount and color of the books should not change as a result of setting them down (*Conservation of Mass*).

Figure 19: Examples of physically unlikely video generations from CogVideoX-5B. Each case demonstrates violations of fundamental physical laws.

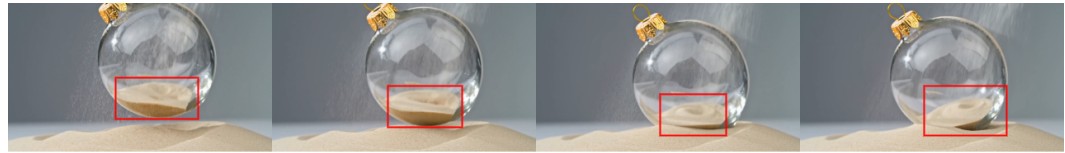

(a) **Caption**: Fine sand is directed at a delicate glass ornament with light pressure, revealing a hidden pattern.
**Human-judged Physical Violations**: The ornament should not collect sand internally if there are no openings, and the sand should not move inside a stationary ornament (*Gravity, Impermeability of Solids*).

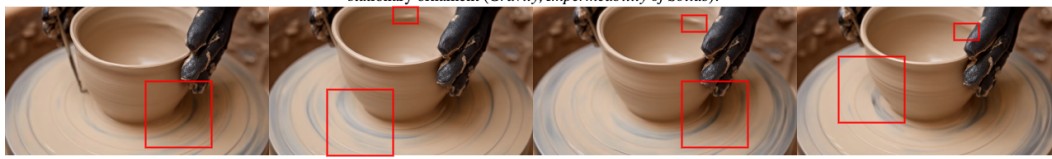

(b) **Caption**: A potter's wheel is lightly poked with a tool, causing a small wobble in the rotating clay.
**Human-judged Physical Violations**: The pot must rotate at the same direction and speed as the base (*Inertia*).

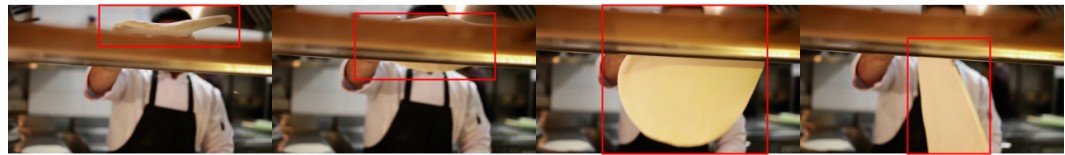

(c) **Caption**: A pizza chef uses a pizza peel to toss and catch a pizza, then slides the pizza onto a baking sheet.
**Human-judged Physical Violations**: The dough's elasticity and hardness should stay constant and not become too hard or soft throughout the video (*Elasticity, Hardness, Gravity*).

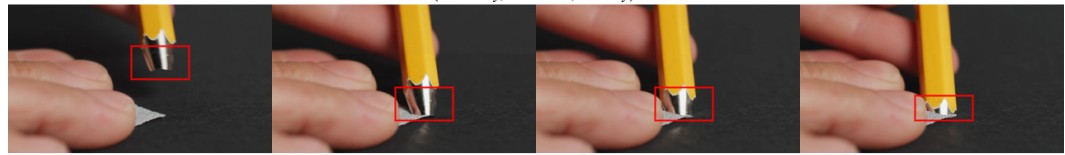

(e) **Caption**: Time-lapse of a pencil being sharpened with sandpaper; the pencil point becomes increasingly sharp.
**Human-judged Physical Violations**: The pencil should maintain its general shape and the mass that it loses should be visible (*Hardness, Conservation of Mass*).

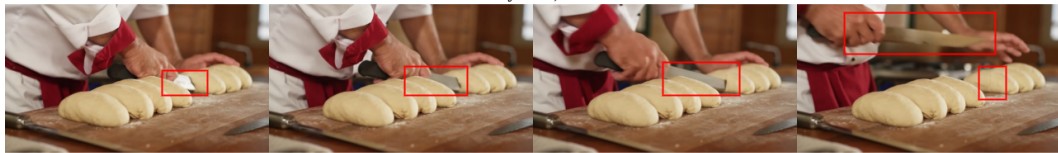

(e) **Caption**: Using a knife, a baker cuts a loaf of pre-shaped dough into smaller rolls, making visible cuts.
**Human-judged Physical Violations**: The knife should protrude out of the roll given its length, and should leave a mark on the dough once removed (*Conservation of Mass, Elasticity, Friction*).

Figure 20: Examples of physically unlikely video generations from Ray2. Each case demonstrates violations of fundamental physical laws.

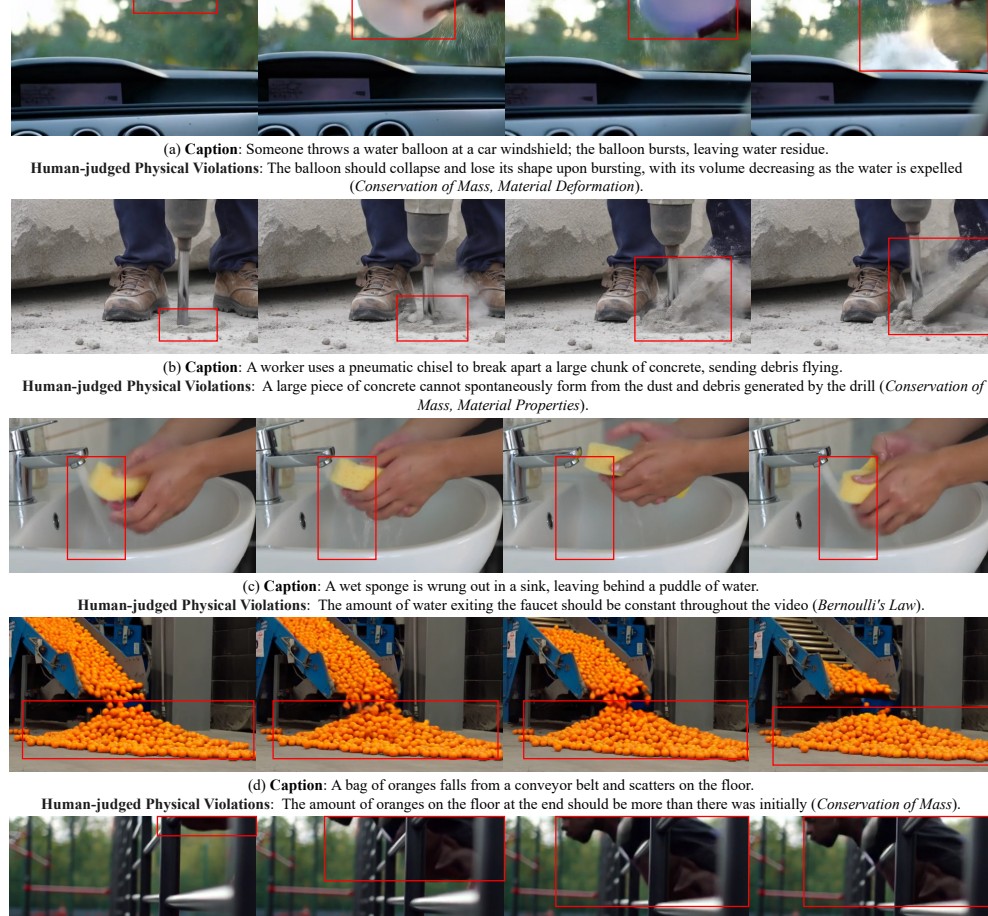

(a) **Caption**: Someone throws a water balloon at a car windshield; the balloon bursts, leaving water residue.
**Human-judged Physical Violations**: The balloon should collapse and lose its shape upon bursting, with its volume decreasing as the water is expelled
(*Conservation of Mass, Material Deformation*).

(b) **Caption**: A worker uses a pneumatic chisel to break apart a large chunk of concrete, sending debris flying.
**Human-judged Physical Violations**: A large piece of concrete cannot spontaneously form from the dust and debris generated by the drill (*Conservation of Mass, Material Properties*).

(c) **Caption**: A wet sponge is wrung out in a sink, leaving behind a puddle of water.
**Human-judged Physical Violations**: The amount of water exiting the faucet should be constant throughout the video (*Bernoulli's Law*).

(d) **Caption**: A bag of oranges falls from a conveyor belt and scatters on the floor.
**Human-judged Physical Violations**: The amount of oranges on the floor at the end should be more than there was initially (*Conservation of Mass*).

(e) **Caption**: Parallel bars are shown from a side view with an athlete performing a series of dips.
**Human-judged Physical Violations**: The athlete's body must make contact with the bars during dips and cannot pass through them (*Impenetrability of Matter, Conservation of Momentum*).

Figure 21: Examples of physically unlikely video generations from Hunyuan. Each case demonstrates violations of fundamental physical laws.

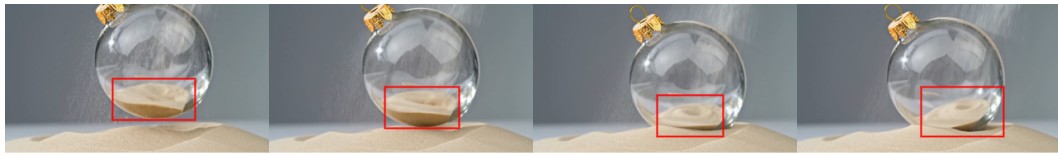

(a) **Caption**: Fine sand is directed at a delicate glass ornament with light pressure, revealing a hidden pattern.
**Human-judged Physical Violations**: The ornament should not collect sand internally if there are no openings, and the sand should not move inside a stationary ornament (*Gravity, Impermeability of Solids*).

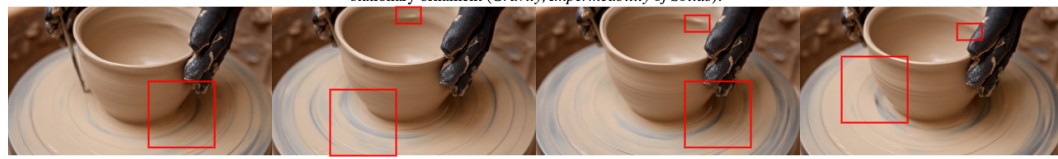

(b) **Caption**: A potter's wheel is lightly poked with a tool, causing a small wobble in the rotating clay.
**Human-judged Physical Violations**: The pot must rotate at the same direction and speed as the base (*Inertia*).

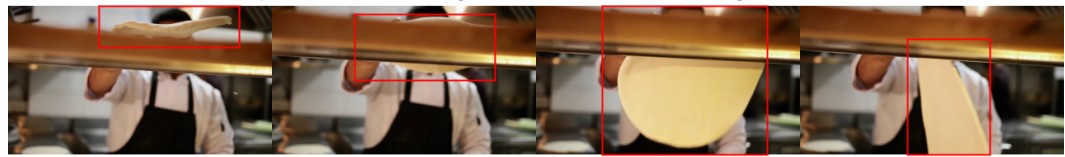

(c) **Caption**: A pizza chef uses a pizza peel to toss and catch a pizza, then slides the pizza onto a baking sheet.
**Human-judged Physical Violations**: The dough's elasticity and hardness should stay constant and not become too hard or soft throughout the video (*Elasticity, Hardness, Gravity*).

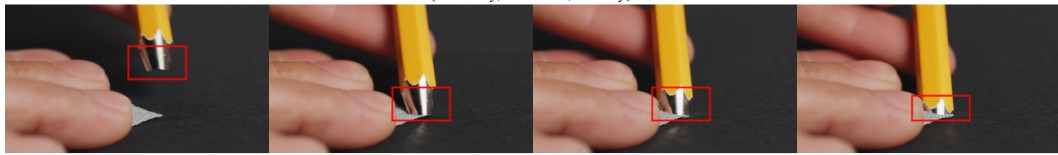

(e) **Caption**: Time-lapse of a pencil being sharpened with sandpaper; the pencil point becomes increasingly sharp.
**Human-judged Physical Violations**: The pencil should maintain its general shape and the mass that it loses should be visible (*Hardness, Conservation of Mass*).

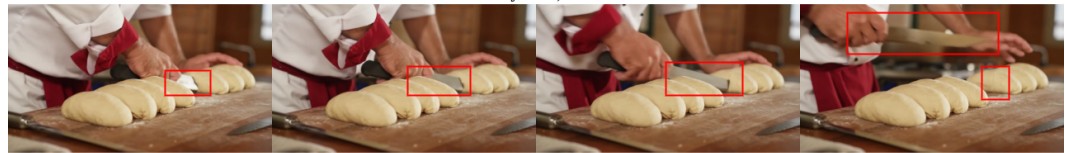

(e) **Caption**: Using a knife, a baker cuts a loaf of pre-shaped dough into smaller rolls, making visible cuts.
**Human-judged Physical Violations**: The knife should protrude out of the roll given its length, and should leave a mark on the dough once removed (*Conservation of Mass, Elasticity, Friction*).

Figure 22: Examples of physically unlikely video generations from Ray2. Each case demonstrates violations of fundamental physical laws.

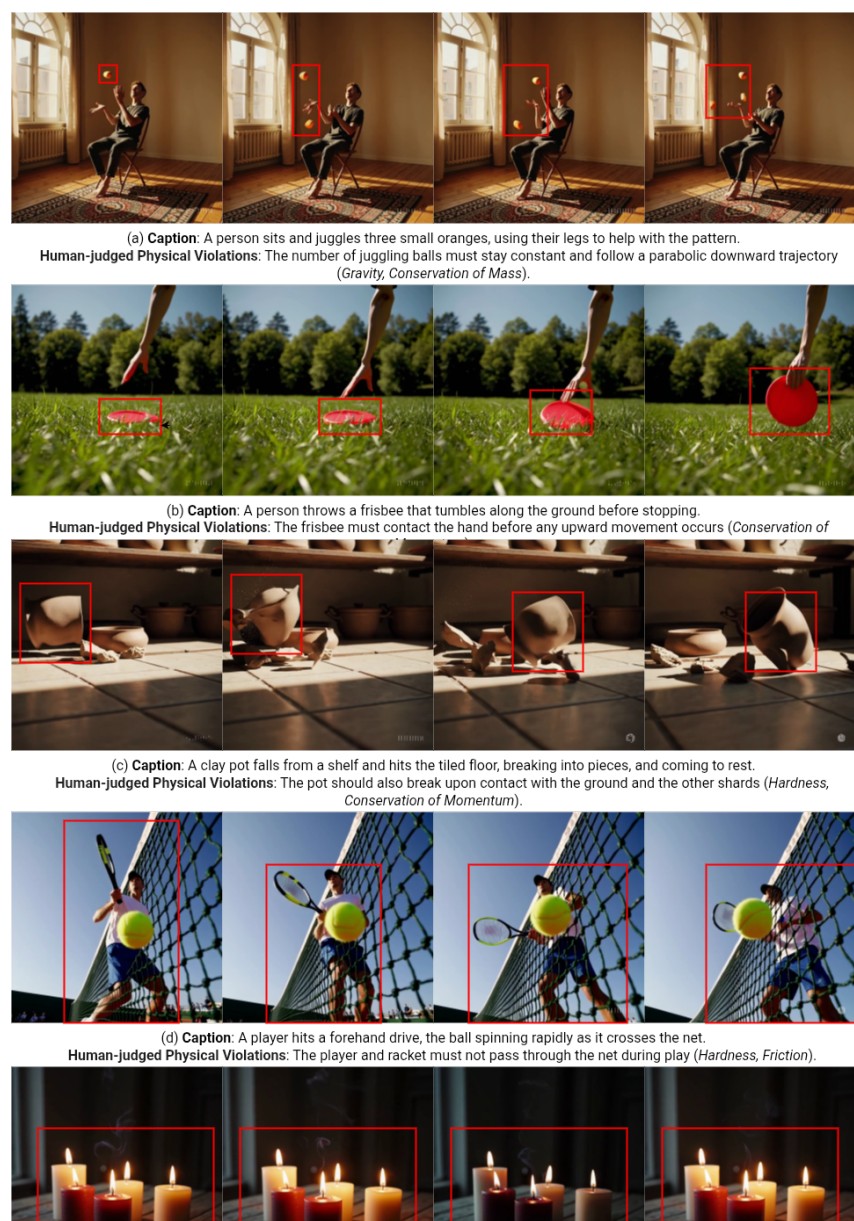

(a) **Caption**: A person sits and juggles three small oranges, using their legs to help with the pattern.
**Human-judged Physical Violations**: The number of juggling balls must stay constant and follow a parabolic downward trajectory (*Gravity, Conservation of Mass*).

(b) **Caption**: A person throws a frisbee that tumbles along the ground before stopping.
**Human-judged Physical Violations**: The frisbee must contact the hand before any upward movement occurs (*Conservation of*

(c) **Caption**: A clay pot falls from a shelf and hits the tiled floor, breaking into pieces, and coming to rest.
**Human-judged Physical Violations**: The pot should also break upon contact with the ground and the other shards (*Hardness, Conservation of Momentum*).

(d) **Caption**: A player hits a forehand drive, the ball spinning rapidly as it crosses the net.
**Human-judged Physical Violations**: The player and racket must not pass through the net during play (*Hardness, Friction*).

(e) **Caption**: Multiple candles of varying heights and widths are blown out simultaneously by a single breath, some flames extinguishing faster than others.
**Human-judged Physical Violations**: If the candles are lit and unwavering, they should emit a constant source of light (*Combustion*).

Figure 23: Examples of physically unlikely video generations from Sora. Each case demonstrates violations of fundamental physical laws.

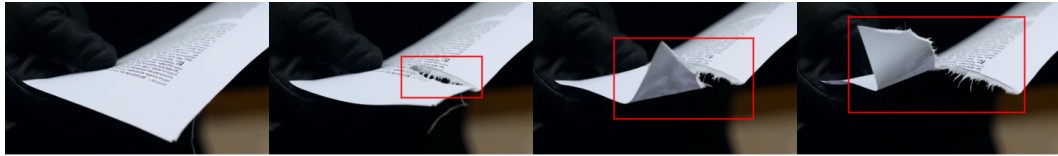

(a) **Caption:** A hand rips a sheet of printer paper in half, creating a jagged tear.
**Human-judged Physical Violations**: The paper requires an external force to initiate the rip (*Newton's Second Law of Motion*).

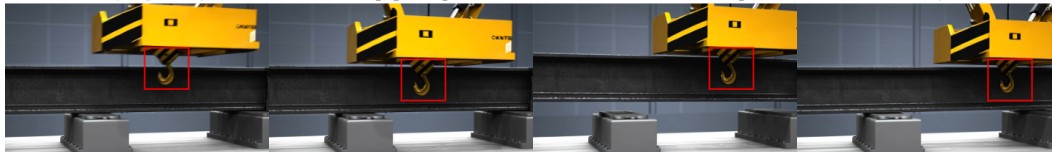

(b) **Caption:** A heavy steel beam, resting on two supports, is lifted at one end by a crane until the beam begins to rotate and slides off the supports.
**Human-judged Physical Violations**: The beam should only be lifted as a result of an upward force exerted by the crane hook that is hooked onto the beam (*Inertia*).

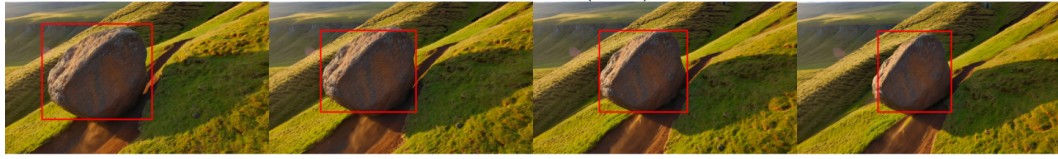

(c) **Caption:** A large stone rolls down a hillside, leaving a visible path in the dirt.
**Human-judged Physical Violations**: The boulder cannot ascend a slope without an external force applied, and the round shape of the rock indicates that it should be rolling (*Gravity, Friction*).

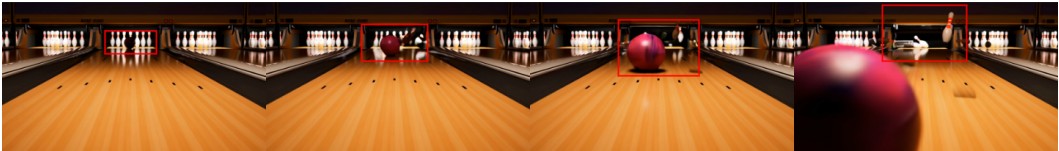

(d) **Caption:** A bowling ball rolls down a polished wooden lane, hitting the pins at the end.
**Human-judged Physical Violations**: The ball starts in front of the pins and should not be able to collide with the pins to push them outwards towards the camera (*Conservation of Momentum*).

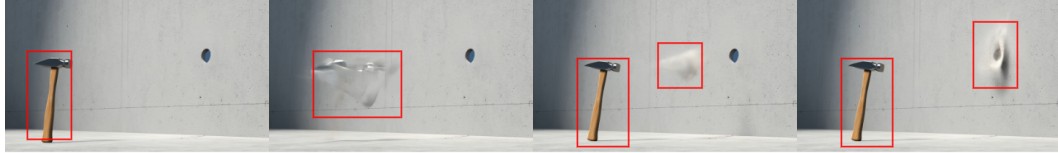

(e) **Caption:** A hammer, thrown with considerable force, bounces off a concrete wall, leaving a visible mark.
**Human-judged Physical Violations**: There needs to be an external force to push the hammer initially, the hammer should not disappear, and there should not be material ejected from the hammer out of nowhere (*Inertia, Material Properties, Conservation of Mass*).

Figure 24: Examples of physically unlikely video generations from Wan2.1. Each case demonstrates violations of fundamental physical laws.

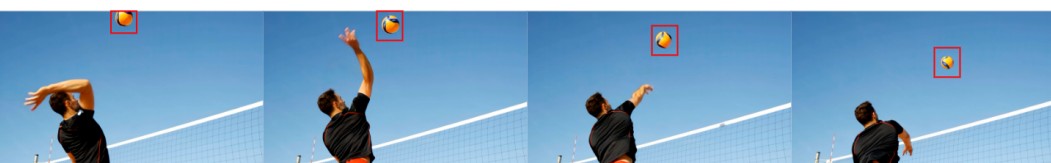

(a) **Caption:** A volleyball player takes a hard swing, the ball contacting the palm and fingers, followed by a rapid downward motion sending the ball over the net.
**Human-judged Physical Violations**: The volleyball cannot move without contact with the player's hand (*Newton's First Law*).

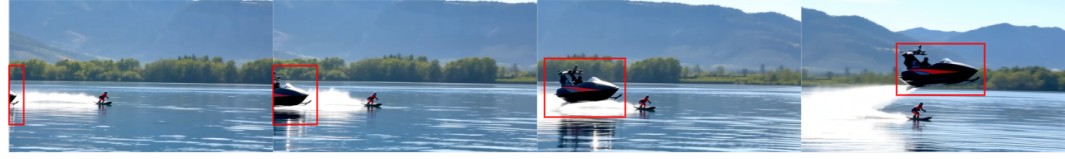

(b) **Caption:** A ski jet pulls a skier, who is towed over the water.
**Human-judged Physical Violations**: The jetski cannot levitate over the water and propulse itself without contact with the water (*Gravity*).

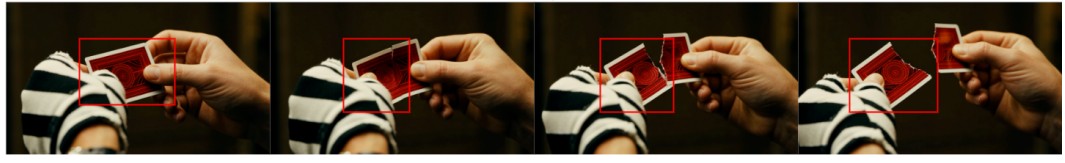

(c) **Caption:** A person rips a playing card in half, separating the two halves cleanly.
**Human-judged Physical Violations**: The playing card cannot gain mass as it is pulled apart (*Law of Conservation of Matter*).

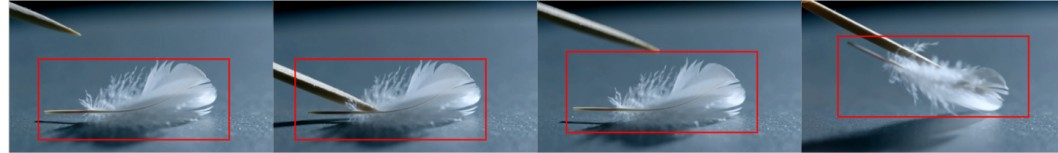

(d) **Caption:** A person rips a playing card in half, separating the two halves cleanly.
**Human-judged Physical Violations**: The playing card cannot gain mass as it is pulled apart (*Law of Conservation of Matter*).

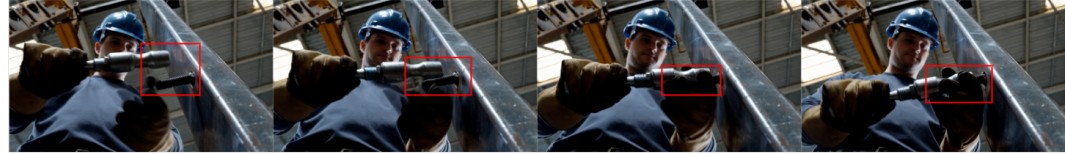

(e) **Caption:** A construction worker uses a socket wrench to turn a large bolt on a metal beam, the wrench exerting significant force.
**Human-judged Physical Violations**: The wrench cannot phase through the bolt (*Pauli's exclusion principle*).

Figure 25: Examples of physically unlikely video generations from Wan2.2. Each case demonstrates violations of fundamental physical laws.

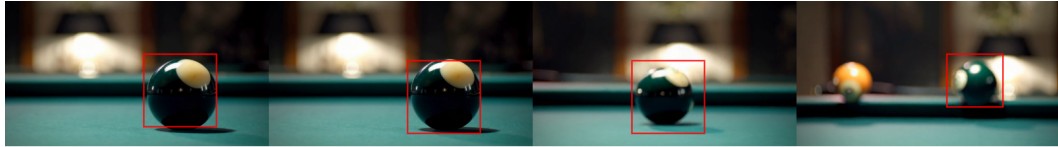

(a) **Caption:** A billiard ball is struck and rolls across a felt-covered pool table, hitting another ball.
**Human-judged Physical Violations**: The billiard ball cannot move and jump into the air without an external force acting on it (*Newton's First Law*).

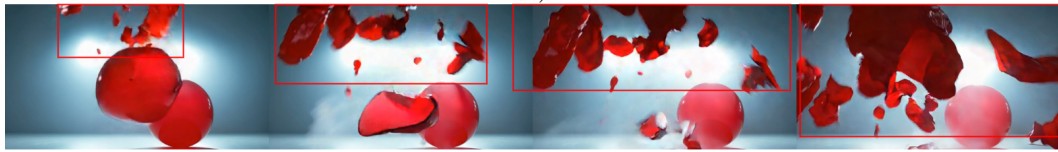

(b) **Caption**: A balloon is popped by being over-inflated, causing it to burst with a loud noise and rapid deflation.
**Human-judged Physical Violations**: The total volume of rubber should remain consistent (*Conservation of Mass*).

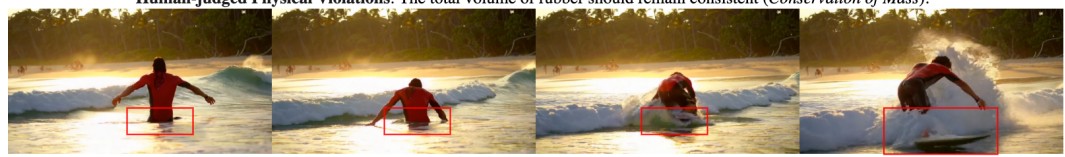

(c) **Caption**: A bowling ball rolls down a polished wooden lane, hitting the pins at the end.
**Human-judged Physical Violations**: The surfer and the board cannot be submerged at the beginning of the wave (*Buoyancy, Conservation of Mass*).

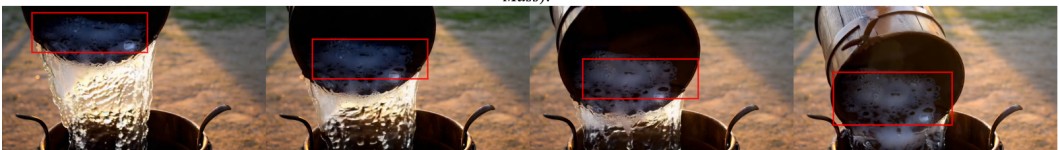

(d) **Caption**: A bucket filled with water tips over, and the water splashes and spills onto the ground.
**Human-judged Physical Violations**: Bubbles on the surface of the water should flow outwards along with the water (*Inertia, Gravity*). Water in the bucket should deplete as water is poured out (*Conservation of Mass*).

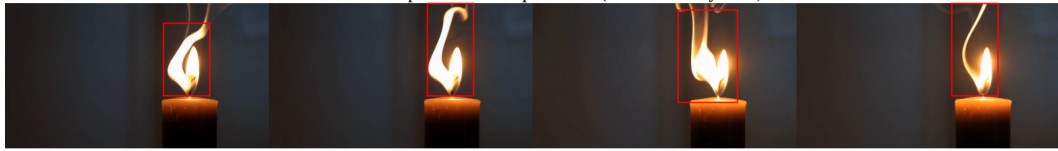

(e) **Caption**: A burning candle produces a small amount of smoke near its wick.
**Human-judged Physical Violations**: The flame should be affected by the same airflow that is affecting the smoke (*Fluid Dynamics*). The smoke should be less illuminated than the fire (*Combustion*).

Figure 26: Examples of physically unlikely video generations from Cosmos-7B. Each case demonstrates violations of fundamental physical laws.

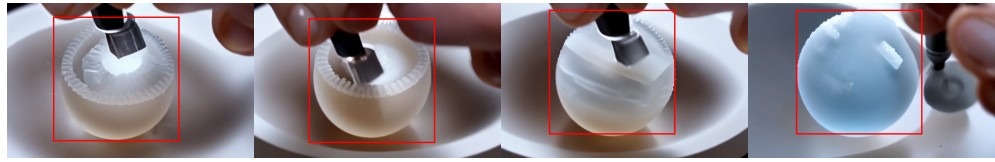

(a) **Caption:** A person carves an ice cube into a perfect sphere using a specialized rotary tool and a steady hand.
**Human-judged Physical Violations**: The tool maintains its shape while carving the ice, and the ices should not increase in volume (*Conservation of Mass*).

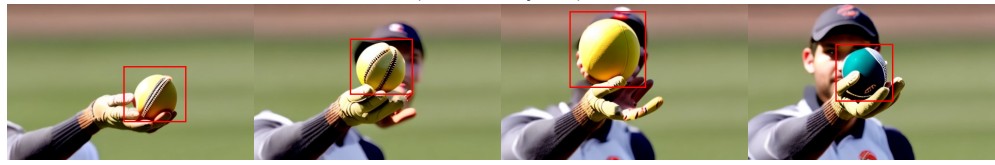

(b) **Caption:** A fielder catches a cricket ball holding it securely in their glove.
**Human-judged Physical Violations**: The ball retains its shape and color during the catch (*Conservation of Mass*).

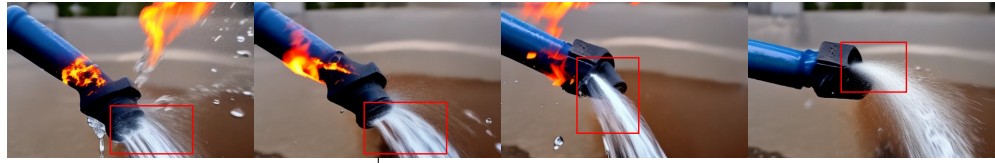

(c) **Caption:** A stream of water from a handheld nozzle hits a small oil fire extinguishing the blaze.
**Human-judged Physical Violations**: The fire should lead a mark and needs to be put out by something, and water should exit the hose at only the nozzle (*Impermeability of Solids, Combustion*)

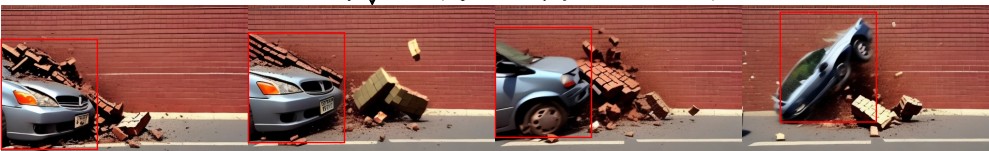

(d) **Caption:** A speeding car crashes into a brick wall crumpling the front end and stopping abruptly.
**Human-judged Physical Violations**: The car cannot shrink or flip upon impact with the wall, and impact duration should be way shorter and sudden (*Friction, Conservation of Mass*).

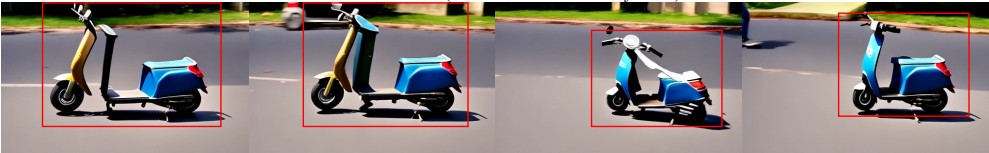

(e) **Caption:** A scooter collides with a trash can the scooter tilting to one side before stopping.
**Human-judged Physical Violations**: The scooter cannot change its model or color spontaneously when in motion (*Conservation of Mass*).

Figure 27: Examples of physically unlikely video generations from VideoCrafter2. Each case demonstrates violations of fundamental physical laws.

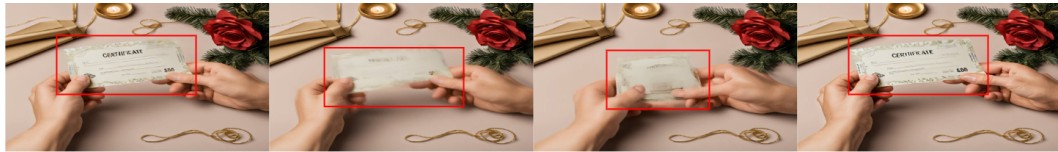

(a) **Caption:** A person folds a gift certificate, placing it into a decorative envelope.
**Human-judged Physical Violations**: The letter should not change its volume (Law of *Conservation of Mass*).

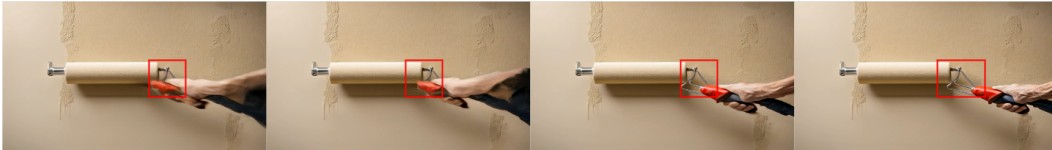

(b) **Caption:** A paint roller applies a coat of beige paint to a textured wall, showing the paint filling in the textures.
**Human-judged Physical Violations**: The paint roller's handle should not bend so easily (*Elasticity, Material Properties*).

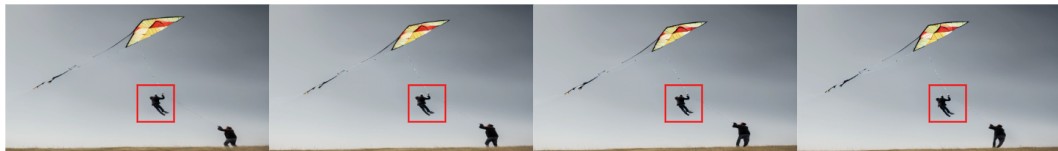

(c) **Caption:** A kite with a camera attached descends gently, lowered by a person pulling the string.
**Human-judged Physical Violations**: The force on the kite and the tension in the string should not be able to hold the person up
(*Gravity*)

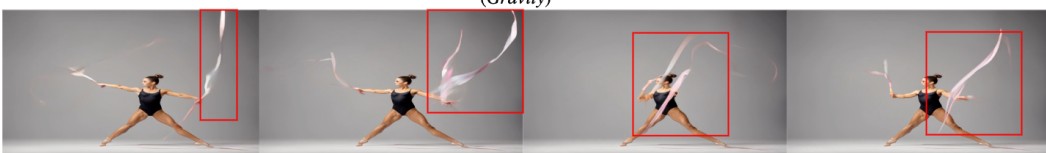

(d) **Caption:** A gymnast stretches a long gymnastic ribbon, extending it into a thin line.
**Human-judged Physical Violations**: The length and width of the ribbon should not change (*Material Properties*).

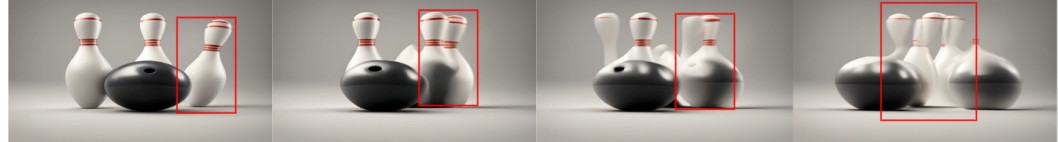

(e) **Caption:** A heavy bowling pin is nudged with the tip of a bowling ball, causing it to fall.
**Human-judged Physical Violations**: The number of bowling pins should stay constant and maintain their shape (*Law of Conservation of Matter*).

Figure 28: Examples of physically unlikely video generations from Stable Video Diffusion. Each case demonstrates violations of fundamental physical laws.

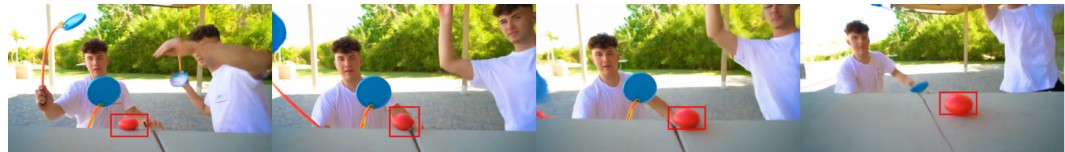

(a) **Model**: Ray2
**Caption**: A person throws a yo-yo, and it strikes a rubber ball, causing the ball to move.
**Human-judged Physical Violations**: The red object should only move as a result of a direct impulse from the paddles. (*Conservation of Momentum*).

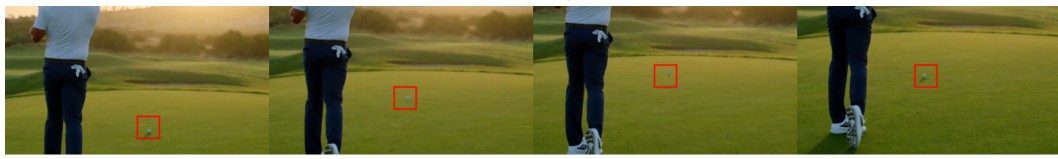

(b) **Model**: Ray2
**Caption**: A golf club strikes a golf ball, sending it rolling across a manicured green toward the hole.
**Human-judged Physical Violations**: The golf ball should only move after being struck by the golf club (*Conservation of Momentum*).

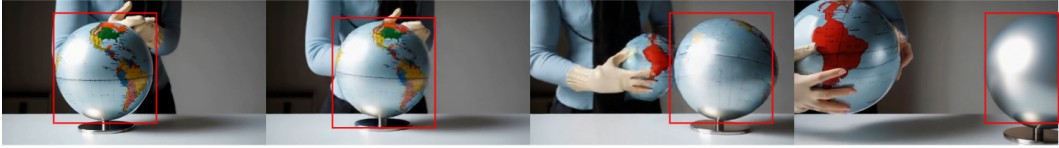

(c) **Model**: Cosmos-1.0
**Caption**: An object shaped like a globe is poked, causing it to spin on its axis, revealing different countries.
**Human-judged Physical Violations**: The globe should only spin and move once it is poked (*Conservation of Momentum*).

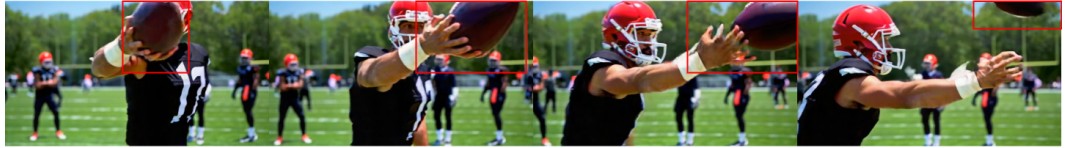

(d) **Model**: Cosmos-1.0
**Caption**: A player throws a sidearm pass; the football travels on a sideways trajectory before being caught.
**Human-judged Physical Violations**: The ball should follow the trajectory dictated by the throw to the right, not upwards (*Conservation of Momentum*).

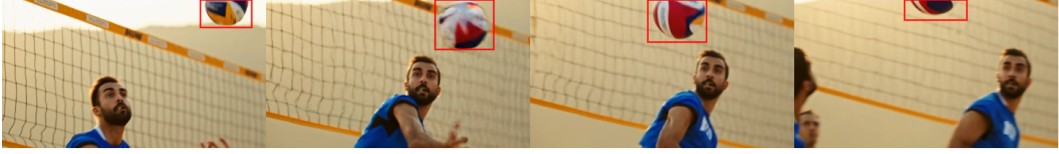

(e) **Model**: Ray2
**Caption**: An outside hitter hits a line shot, the ball bouncing sharply off the side line.
**Human-judged Physical Violations**: The ball should only change direction after colliding with anothe robject (*Conservation of Momentum*).

Figure 29: Examples of physically unlikely video generations where the physical law of conservation of momentum is violated

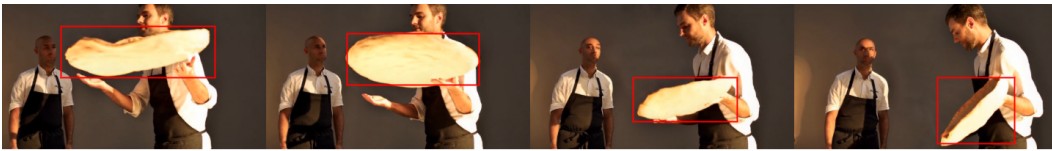

(a) **Model**: Cosmos-1.0
**Caption**: A pizza dough is tossed up, its edges become thinner as it rotates in the air, and it's caught by a second person.
**Human-judged Physical Violations**: The pizza dough maintains its volume while stretching during the toss (*Conservation of Mass*).

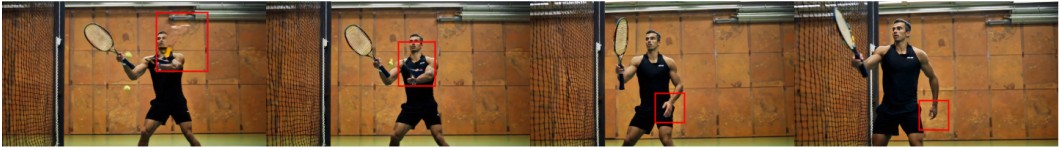

(b) **Model**: Cosmos-1.0
**Caption**: A squash player serves the ball, the ball striking the front wall with a visible impact.
**Human-judged Physical Violations**: The player's left hand should not hold an object that disappears (*Conservation of Mass*).

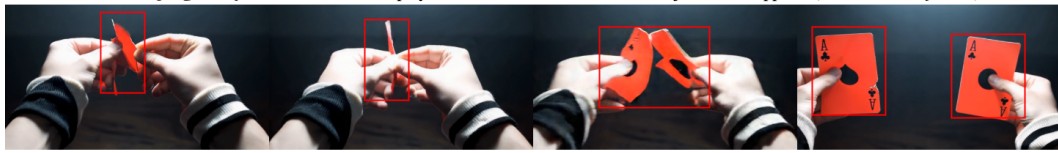

(c) **Model**: Cosmos-1.0
**Caption**: A person rips a playing card in half, separating the two halves cleanly.
**Human-judged Physical Violations**: The total amount of paper should remain constant and not appear out of nowhere (*Conservation of Mass*).

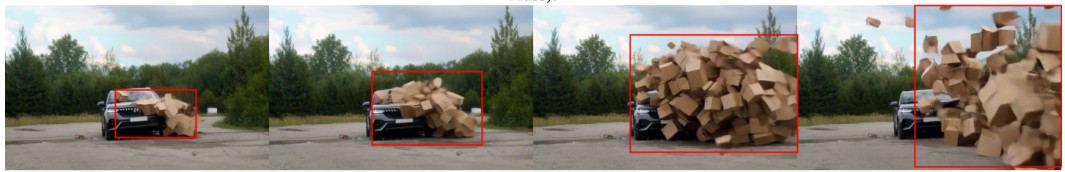

(d) **Model**: Hunyuan
**Caption**: A car crashes into a stack of cardboard boxes, sending the boxes flying in all directions.
**Human-judged Physical Violations**: The total number of boxes remains constant before and after the impact (*Conservation of Mass*).

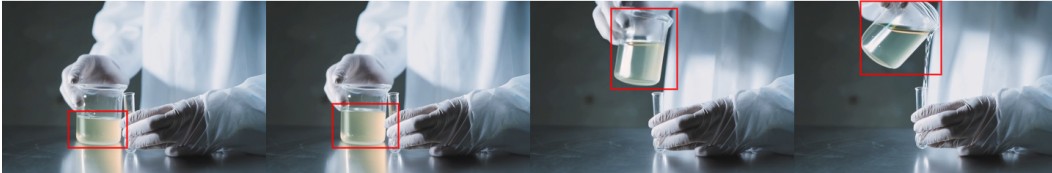

(e) **Model**: CogvideoX-5B
**Caption**: A chemist pours a clear liquid from a beaker into a test tube, carefully avoiding spills.
**Human-judged Physical Violations**: The liquid level does not rise in the beaker during pouring and the height of the beaker should not increase (*Conservation of Mass*).

Figure 30: Examples of physically unlikely video generations where the physical law of conservation of mass is violated

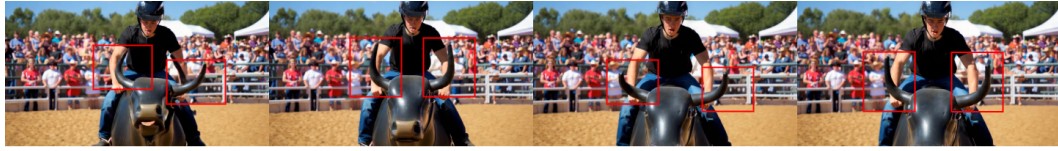

(a) **Model**: Cosmos-1.0
**Caption**: A rider sits on the mechanical bull, which starts to buck gently; then, the intensity gradually increases.
**Human-judged Physical Violations**: The bull's metal horns should not deform nor return to their original positions (*Elasticity*).

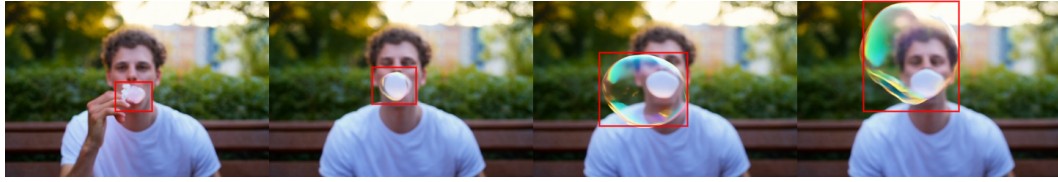

(b) **Caption**: A person blows a bubble gum bubble while sitting on a park bench, the bubble floating away.
**Human-judged Physical Violations**: The bubble should not be able to stretch so think that it becomes as thin and transparent as a soap bubble (*Elasticity*).

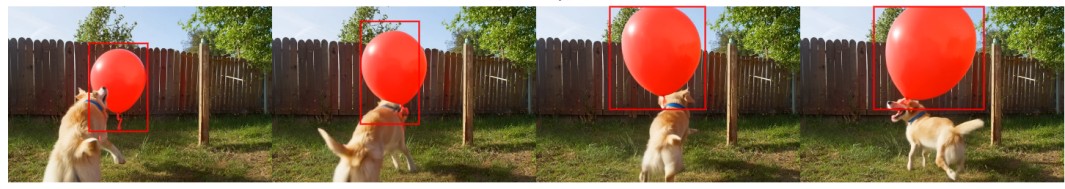

(c) **Model**: CogvideoX-5B
**Caption**: A dog playfully bats at a balloon, causing it to pop against a nearby fence.
**Human-judged Physical Violations**: The balloon should not expand as the dog plays with it (*Elasticity*).

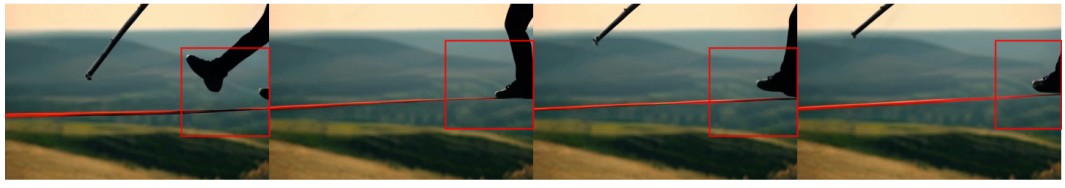

(d) **Model**: CogvideoX-5B
**Caption**: A tightrope walker, with their balance pole extended to the side, uses it to brace against a sudden gust of wind.
**Human-judged Physical Violations**: The rope should bend or deform in response to the person shifting their weight with their feet (*Elasticity*).

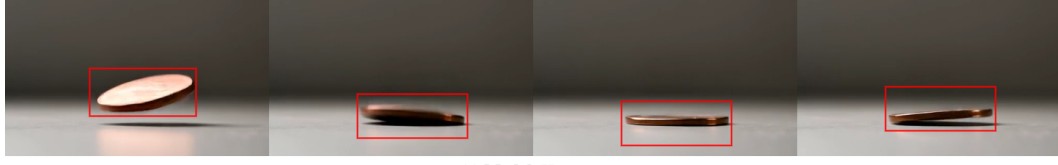

(e) **Model**: Hunyuan
**Caption**: A coin spins on a flat surface, coming to rest on heads.
**Human-judged Physical Violations**: The coin does not bounce off the surface after coming to rest (*Elasticity*).

Figure 31: Examples of physically unlikely video generations where the physical law of elasticity is violated

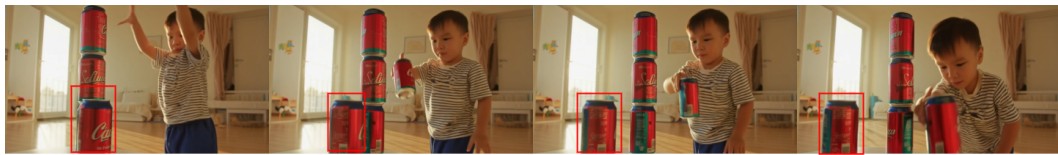

(a) **Model**: Ray2
**Caption**: A small child pokes a stack of empty soda cans with a toy car, causing the cans to topple like a chain reaction.
**Human-judged Physical Violations**: The cans stop moving and rotating due to friction (*Friction*).

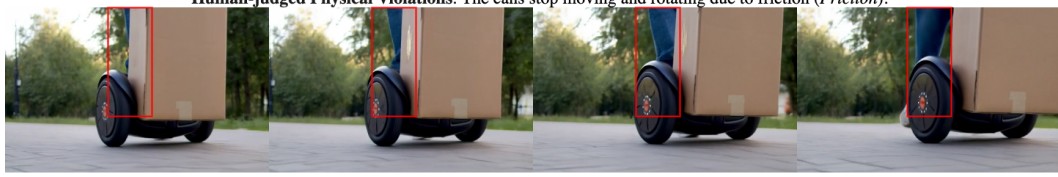

(b) **Model**: Hunyuan
**Caption**: A package is secured to a Segway's platform; the Segway travels a short distance, delivering the package.
**Human-judged Physical Violations**: The person's foot should not move relative to the segway (*Friction*).

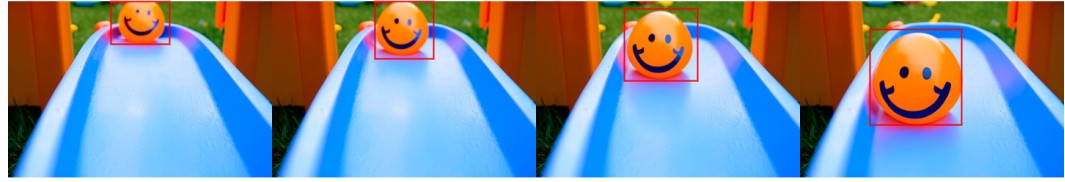

(c) **Model**: CogvideoX-5B
**Caption**: A large, orange plastic egg is rolled up a plastic slide, then rolls down, ending at the bottom.
**Human-judged Physical Violations**: The ball should roll down the slide with rotational motion. (*Friction*).

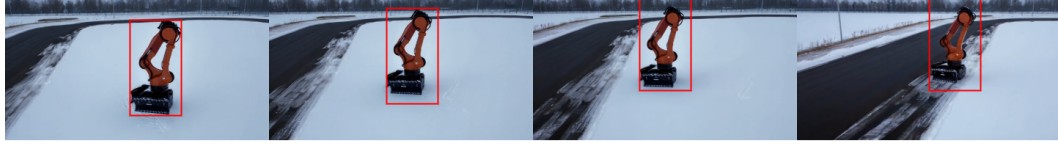

(d) **Model**: Cosmos-1.0
**Caption**: A robotic arm uses a shovel to clear snow from a large parking lot.
**Human-judged Physical Violations**: The device should leave a visible track as it slides through the snow. (*Friction*).

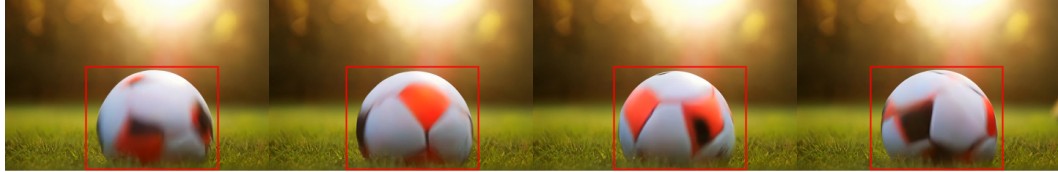

(e) **Model**: CogvideoX-5B
**Caption**: A soccer ball is kicked and hits a person's leg, deflecting at a different angle.
**Human-judged Physical Violations**:The ball should experience an acceleration in speed as it spins in with the grass (*Friction*).

Figure 32: Examples of physically unlikely video generations where the physical law of friction is violated

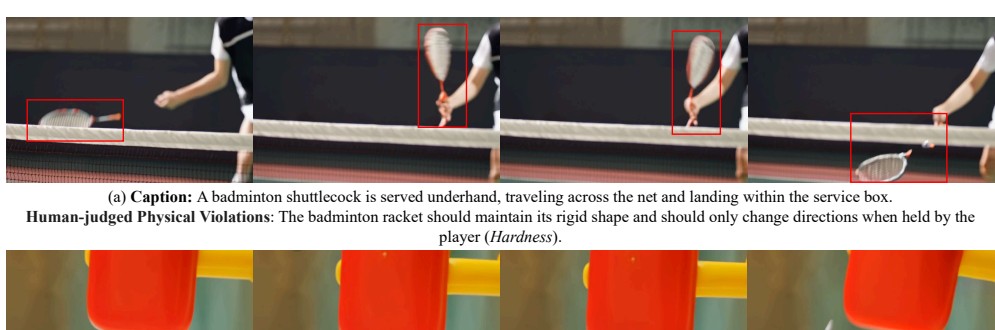

(a) **Caption:** A badminton shuttlecock is served underhand, traveling across the net and landing within the service box.
**Human-judged Physical Violations**: The badminton racket should maintain its rigid shape and should only change directions when held by the player (*Hardness*).

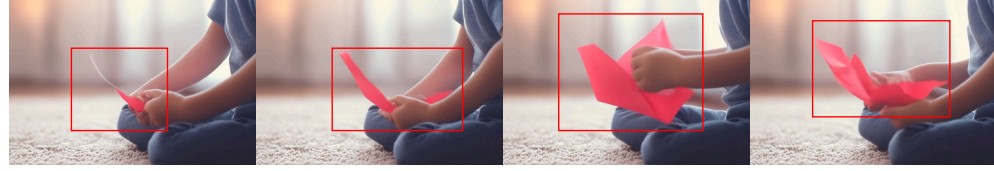

(b) **Caption:** A child's toy hammer smashes a small plastic egg, breaking it open.
**Human-judged Physical Violations**: The mass and amount of shards of egg should stay constant, no mass should appear out of nowhere (*Conservation of Mass*).

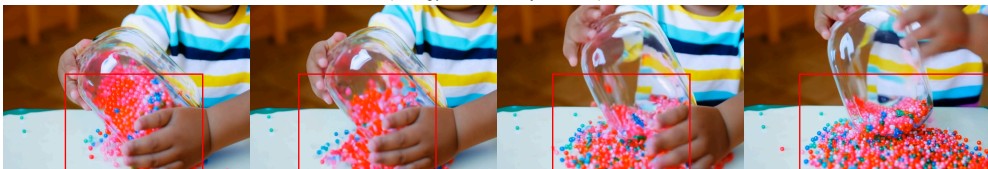

(c) **Caption:** A child folds a piece of origami paper into a simple crane, with visible creases appearing.
**Human-judged Physical Violations**: Creases should not spontaneously form in the paper without an external force or proper folding sequence (*Entropy, Material Deformation*).

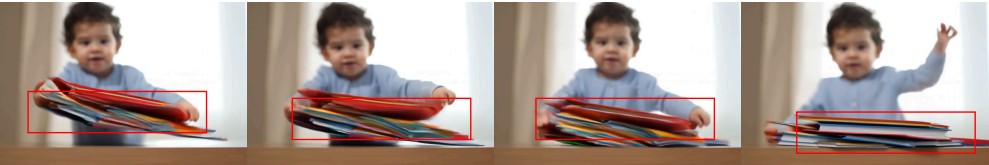

(d) **Caption:** A child pours colorful beads from a plastic container into a glass jar until they overflow, scattering on the floor.
**Human-judged Physical Violations**: The container should not leak beads with no openings (*Gravity, Impermeability of Solids*).

(e) **Caption:** A child pushes a stack of books off a desk; the books fall in a chaotic pile.
**Human-judged Physical Violations**: The amount and color of the books should not change as a result of setting them down (*Conservation of Mass*).

Figure 33: Examples of physically unlikely video generations where the physical law of gravity is violated

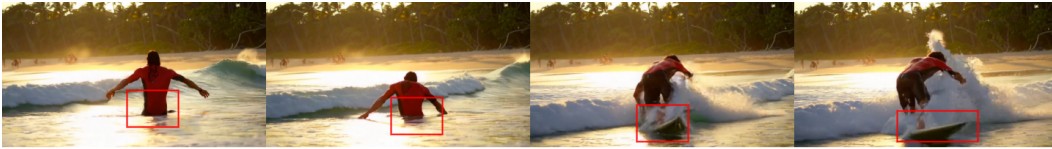

(a) **Model**: Cosmos-1.0
**Caption**: A bodysurfer dives headfirst into a wave, emerging from the whitewater several seconds later.
**Human-judged Physical Violations**: The surfer should not be able to stay submerged during the first part of the video due to the buoyancy from their board (*Buoyancy*).

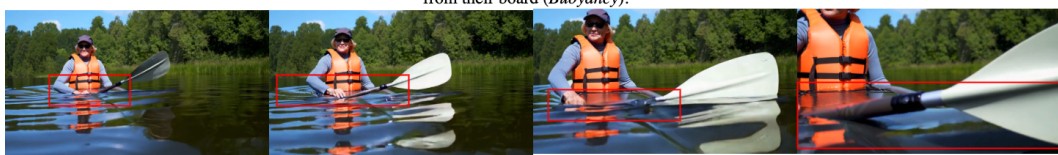

(b) **Model**: Cosmos-1.0
**Caption**: A kayaker lifts their paddle out of the water, showing the surface tension of the water clinging to the blade.
**Human-judged Physical Violations**: The woman is too deep in the water such that the kayak should be sinking (*Buoyancy*).

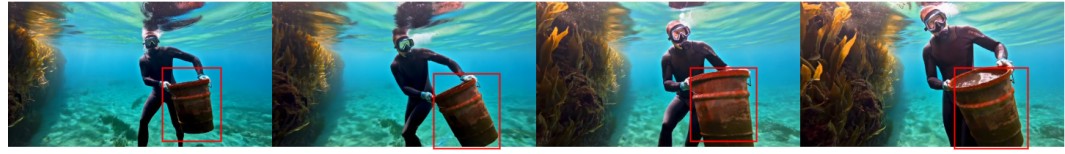

(c) **Model**: Cosmos-1.0
**Caption**: A person carries a heavy bucket while wading through chest-deep water.
**Human-judged Physical Violations**: The bucket should not have another liquid inside of it and if it did have a denser liquid it would be much harder to carry (*Buoyancy*).

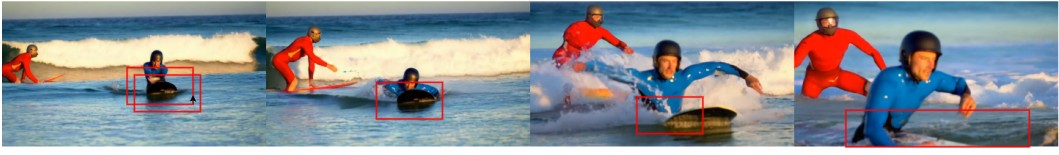

(d) **Model**: Cosmos-1.0
**Caption**: A surfer falls off their board, colliding briefly with another surfer, before swimming away.
**Human-judged Physical Violations**: The surfer's legs should remain above the water surface when not on the board (*Buoyancy*).

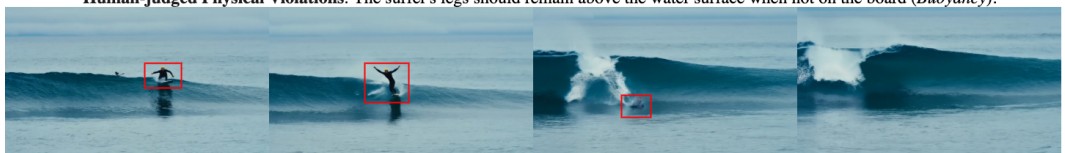

(e) **Model**: Ray2
**Caption**: A surfer dives underwater to avoid a breaking wave, resurfacing seconds later.
**Human-judged Physical Violations**: The surfer should create a splash when entering the water and reappear shortly after (*Buoyancy*).

Figure 34: Examples of physically unlikely video generations where the physical law of Buoyancy is violated

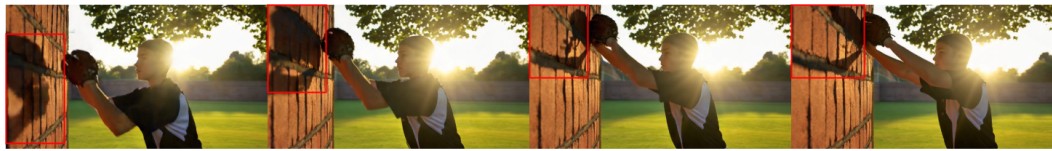

(a) **Model**: Cosmos-1.0
**Caption**: A player throws a softball against a brick wall; the ball rebounds at a visible angle.
**Human-judged Physical Violations**: The shadow on the brick wall should exhibit the outline of both of the player's arms with no distortions (*Reflection*).

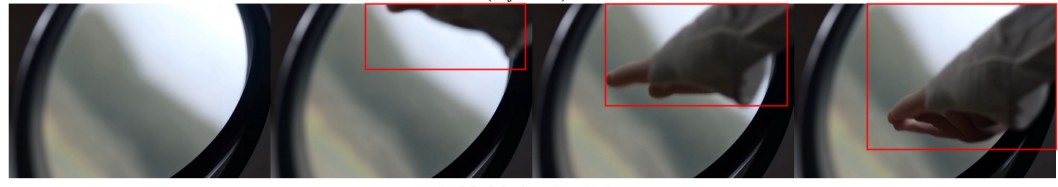

(b) **Model**: CogvideoX-5B
**Caption**: A large, circular mirror is slightly poked, causing a minor shift in its reflected image, though it does not spin.
**Human-judged Physical Violations**: The hand's reflection should be visible in the mirror when it is in front of the surface (*Reflection*).

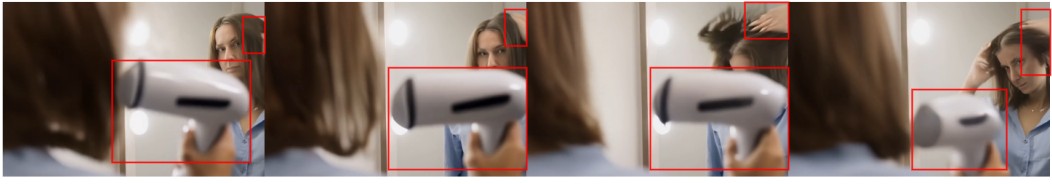

(c) **Model**: CogvideoX-5B
**Caption**: A person uses a low heat setting on the hairdryer to gently dry their fine hair.
**Human-judged Physical Violations**: The hairdryer should be visible in the mirror's reflection (*Reflection*).

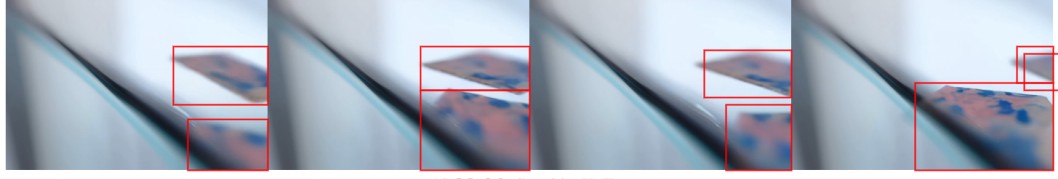

(d) **Model**: CogvideoX-5B
**Caption**: A small, rectangular piece of cardboard is placed on a sloped glass surface; it slides down slowly, leaving no visible trace.
**Human-judged Physical Violations**: The cardboard should leave a reflection on the glass that matches its orientation consistently (*Reflection*).

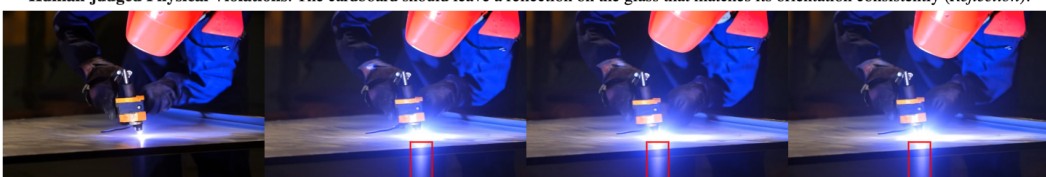

(e) **Model**: CogvideoX-5B
**Caption**: A welder uses a plasma cutter to cut a steel sheet, producing a bright arc and a clean cut edge.
**Human-judged Physical Violations**: Light should not pass through the steel sheet and reflect off of the air into the camera (*Reflection*).

Figure 35: Examples of physically unlikely video generations where the physical law of reflection is violated

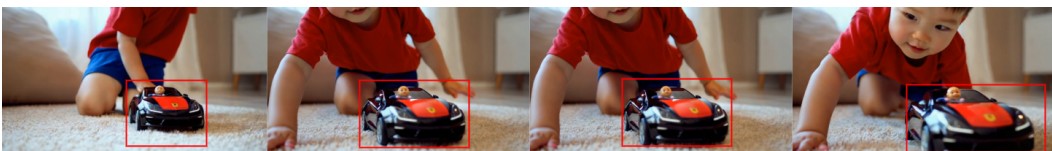

(a) **Model**: Cosmos-1.0
**Caption**: A child pushes a toy car across a carpeted floor; the car rolls a short distance before stopping.
**Human-judged Physical Violations**: The car should not move after it stops, unless pushed again by an external force (*Inertia*).

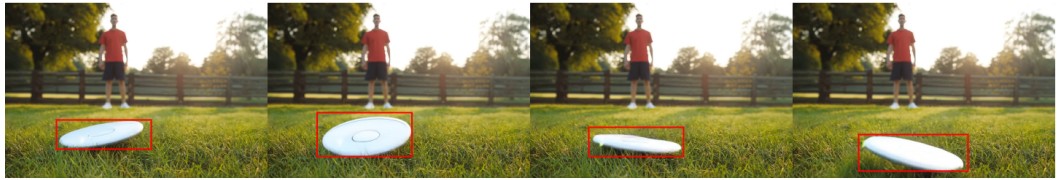

(b) **Model**: Ray2
**Caption**: Concrete is poured from a wheelbarrow into a form, leveling the surface with a trowel.
**Human-judged Physical Violations**: The wheelbarrow should not move since the man is not touching it (*Inertia*).

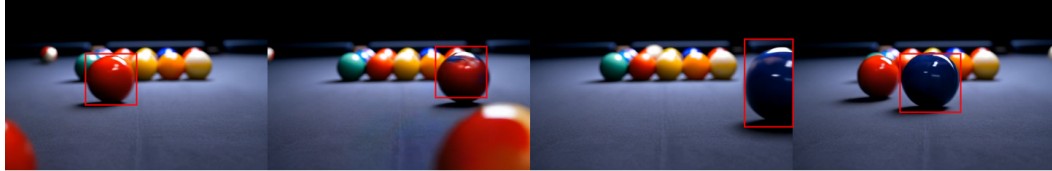

(c) **Model**: CogvideoX-5B
**Caption**: A person throws a frisbee that tumbles along the ground before stopping.
**Human-judged Physical Violations**: The frisbee should not move without an external force acting on it (*Inertia*).

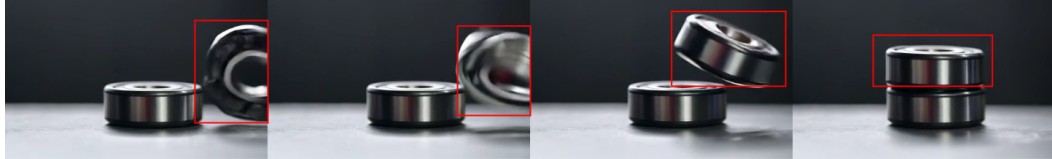

(d) **Model**: CogvideoX-5B
**Caption**: Multiple balls are grouped near a pocket; a precise shot pockets two of the balls.
**Human-judged Physical Violations**: The frisbee should not move without an external force acting on it (*Inertia*).

(e) **Model**: Hunyuan
**Caption**: Multiple balls are grouped near a pocket; a precise shot pockets two of the balls.
**Human-judged Physical Violations**: The bearing should change direction only due to an applied force or interaction with another object and should only stop rotation as a result of an external torque (*Inertia*).

Figure 36: Examples of physically unlikely video generations where the physical law of inertia is violated