# OpenReview forum: "VideoPhy-2: A Challenging Action-Centric Physical Commonsense Evaluation in Video Generation"
_ICLR.cc/2026/Conference — ICLR 2026 Poster_

### Official Review · Reviewer_tmU3 · 2025-10-30

**Soundness:** 4
**Presentation:** 3
**Contribution:** 3
**Rating:** 6
**Confidence:** 3

**Summary:**

Video generation models can produce realistic videos, making them ideal candidates for simulating the physical world. However, evaluating how well these models adhere to physical commonsense in real-world actions remains unclear. Existing models face issues such as limited size and lack of human evaluation, making it difficult to assess their ability to align with physical laws. In this work, the authors propose the Videophy-2 benchmark, which includes a dataset of 4000 diverse and detailed prompts, along with an automatic evaluator. Experimental results show that, under the Videophy-2 standard, even the best model achieves only 47.7% joint performance, exposing critical gaps in video generative models and providing direction for future research in physically-grounded video generation.

**Strengths:**

- It provides a challenging benchmark for physically-grounded video generation.

In the hard subset of Videophy-2, even the highest-performing model, Wan2.2-27B-A14B, only achieves a score of 47.7%. This means that Videophy-2 is a stringent benchmark, setting a standard for future evaluations of video generation models in terms of physical alignment.

- Detailed evaluation dimensions.

 Unlike existing work that combines semantic and physical evaluations, leading to potential biases, Videophy-2 uses three distinct dimensions: Semantic Adherence (SA), Physical Commonsense (PC), and Physical Rules (PR). These dimensions evaluate whether the video aligns with the prompt, whether the video is physically aligned with the real world, and whether the video follows a specific physical law (e.g., gravity). This approach minimizes evaluation biases as much as possible.

- An effective automated evaluation approach.

Although human judgments can serve as the gold standard, the cost of human annotation limits the widespread application of evaluations. Therefore, Videophy-2 introduces an automatic evaluation method, using the VideoCon-Physics fine-tuned Videophy-2-autoeval (7B parameters) as the annotator. Experiments show that Pearson’s correlation of Videophy-2-autoeval exceeds that of several VLMs, including Gemini-2.0-Flash-Exp, and surpasses VideoCon-Physics in F1 score.

**Weaknesses:**

- The lack of comparison with more advanced VLMs as annotators.

The most advanced closed-source VLM used in this paper is Gemini-2.0-Flash-Exp, but there is no comparison with stronger models like Gemini-2.5. Given the relatively small evaluation dataset, using stronger VLMs as annotators would increase the credibility of the conclusions drawn in the paper.

**Questions:**

- Is it possible to prove that Videophy-2-autoeval does not suffer from evaluation bias?

In this paper, the authors collect labeled data sampled from HunyuanVideo-13B, Cosmos-Diffusion-7B, and CogVideoX-5B as the training set for Videophy-2-autoeval and evaluate its performance on unseen prompts and unseen video models. However, the videos generated by HunyuanVideo-13B, Cosmos-Diffusion-7B, and CogVideoX-5B may contain biases in terms of adhering to or violating physical laws. Given that the annotator is trained on data generated by these models, can it be considered a fair standard for evaluating future video models?

- Would incorporating real videos into the training set of Videophy-2-autoeval improve its evaluation performance?

The training set of Videophy-2-autoeval only includes synthetic data, whereas real videos inherently possess the best physical commonsense. It is a question that adding real videos to the training data and constructing a diverse training set by providing different prompt-video and rule-video pairs could help make Videophy-2-autoeval closer to human annotators.

- Can I2V models be evaluated using VideoPhy-2?

In this paper, the authors mainly compare the performance of text-to-video (T2V) models in terms of physical alignment. For image-to-video (I2V) models, the initial image should significantly influence the evaluation score. The authors evaluate the performance of the SVD-I2V model by first generating an image using Stable Diffusion and then guiding the SVD-I2V model to generate a video based on that image. Is there a better method to assess the physical alignment performance of I2V models?

---

> ### Author Response · Authors · 2025-11-16
> **Rebuttal**
>
> We thank the reviewer for their feedback. We are excited to see that the reviewer finds our work: (a) is a stringent benchmark that sets a standard for future evaluations, (b) minimizes evaluation biases, and (c) presents an effective automatic evaluation approach.
>
> **Q: Comparison with more advanced models**
>
> - We thank the reviewer for suggesting this experiment. Based on their feedback, we got semantic adherence (SA) and physical commonsense (PC) judgments from Gemini-2.5-Flash. We provide the correlation between the ground-truth SA and PC with the predictions below:
> | Method                                       | Avg   | SA     | PC    |
> |----------------------------------------------|-------|--------|-------|
> | Gemini-Flash-2                               | 18.5  | 26     | 11    |
> | Gemini-Flash-2.5                             | 20.5  | 31     | 10    |
> | VideoPhy2-AutoEval                           | 42    | 47     | 37    |
> | Rel. Improvement in comparison to Gemini-2.5 | **+104%** | **+51.6%** | **+270%** |
> - Interestingly, we find that VideoPhy2-AutoEval performs much better than Gemini-Flash-2.5, consistent with our findings for Gemini-Flash-2.0. This suggests that recent improvements in state-of-the-art VLMs are not focused on enhancing semantic adherence or physical commonsense judgments for synthetic videos. Therefore, using our open and free AutoEval remains the more reliable choice for the future.
>
> **Q: Evaluation Bias in VideoPhy2-AutoEval**
>
> - Firstly, we highlight that VideoPhy2-AutoEval is finetuned on a large set of human annotations collected from three diverse video models. In Tables 4 and 5, we show that VideoPhy2-AutoEval’s correlation and accuracy on unseen video models (beyond Hunyuan, Cosmos, and CogVideoX) remain significantly higher than all baselines. This is largely because most baselines have never been exposed to high-quality data capturing semantic adherence and physical commonsense, as in our work.
> - We do observe a slight drop in performance on unseen videos (i.e., new prompts and unseen video models) compared to unseen prompts (i.e., new prompts but seen video models), but this difference is not substantial. Therefore, as new video generative models emerge, we expect VideoPhy2-AutoEval to maintain relatively high correlation with human annotations, much higher than any other expensive baseline (e.g., Gemini) available on our dataset.
> - Additionally, it is important to note that VideoPhy2-AutoEval is built on top of a VideoLLM foundation model that has been trained on internet-scale video data. Consequently, it can generalize to a much broader distribution than the three models used during finetuning.
> - Overall, we believe that developing strong models for physical understanding is an active and evolving research area, and VideoPhy2 provides an open and solid platform to achieve this goal.
>
> **Q: Clarification on real videos into training set of VideoPhy-2-autoeval**
> - We respectfully clarify that our evaluation focuses on semantic adherence, physical commonsense, and physical rules in synthetic videos, not real videos. Therefore, it is essential to expose models to synthetic data distributions to achieve strong performance in this setting.
> - We also highlight that strong models like Gemini are likely trained extensively on real videos, yet they still do not outperform our AutoEval. This is due to the substantial distribution gap between synthetic and real videos, along with the inherent difficulty of the tasks in VideoPhy2.
> - While real videos naturally encode ground-truth physical commonsense, our AutoEval already benefits from exposure to large-scale real video data during its pretraining. As a result, any physical understanding that can be learned from real videos is already embedded in the foundation model.
> - In addition, the question of how much physical commonsense can be learned from real videos alone remains an open research problem that requires separate investigation. Notably, even current video generative models, despite being trained on internet-scale real video corpora, perform poorly on VideoPhy2 prompts. This suggests that learning real-world physics remains challenging at existing model capacities and training data scales.

---

> > ### Author Response · Authors · 2025-11-16
> > **Rebuttal continued**
> >
> > **Q: I2V models**
> >
> > - We respectfully highlight that the scope of the VideoPhy-series benchmarks is to evaluate the ability of video models to generate physically realistic and accurate synthetic videos from text prompts. This design choice is motivated by the natural way users communicate intent to generative models, for example by describing novel scenes or objects in natural language.
> > - In our experiments, we evaluate the SVD-I2V model using its default and recommended usage. We fully acknowledge that the initial image strongly influences all future frames in I2V models, but this is an inherent property of the I2V paradigm and cannot be circumvented.
> > - In Lines 76–79, we already discuss these challenges when comparing the Physics-IQ benchmark, which is centered around I2V models. Physics-IQ conditions video models on the first few frames of real videos and evaluates quality by comparing predicted videos with the ground-truth completions. However, this setup faces several limitations: (a) the degree to which it aligns with human judgment is unclear, and (b) extending it to complex scenarios involving multiple events is non-trivial.
> > - That said, we take an optimistic view and believe that targeted research is needed to better understand the behavior of I2V models in multi-event scenarios, especially when the initial frame shows only a single event. We will highlight this nuance in the camera-ready version.

---

> > > ### Author Response · Authors · 2025-11-28
> > > **Reviewer Reminder**
> > >
> > > Hi Reviewer tmU3,
> > >
> > > Thank you again for handling our paper. Since the author–reviewer discussion period is coming to an end, could you please let us know if there are any further questions?

---

### Official Review · Reviewer_74jg · 2025-10-30

**Soundness:** 2
**Presentation:** 2
**Contribution:** 2
**Rating:** 4
**Confidence:** 3

**Summary:**

VIDEOPHY-2 is a large-scale benchmark for evaluating the physical common sense of video generation, centered on real-world "actions". It includes 3940 detailed prompts, multi-human evaluation, and fine-grained physics rule annotations, and provides an automated evaluator. Experiments show that even the strongest model achieves only about 47.7% of the joint semantic and physical metrics on hard sets, particularly weak in conservation laws (mass, momentum), highlighting the significant shortcomings of current video generation in terms of physical consistency.

**Strengths:**

The benchmark is based on 197 real-world actions and 3940 long text prompts, covering diverse scenarios that closely resemble everyday physical situations.

It includes human-annotated semantic and physical scores, along with compliance and violation annotations at the level of physical rules/laws, supporting detailed diagnostics.

It provides an AUTOEVAL model to improve evaluation efficiency and consistency with human judgment, facilitating large-scale, rapid model comparisons.

**Weaknesses:**

This paper has limited noverty, and the only difference between it and VideoPhy seems to be the addition of more prompts; it is not suitable as a standalone conference paper.

Table 5 tests the OOD generalization ability of the eval model, but lacks comparisons using some cutting-edge VLMs, such as Qwen and GPT.

The paper lacks insight into how to enhance the ability of T2V models to generate physically consistent videos. Several necessary experiments could provide further insights, such as SFT, using an Eval model as a reward model for RL training, or RFT.

**Questions:**

see the weaknesses

---

> ### Author Response · Authors · 2025-11-16
> **Author Rebuttal**
>
> We are happy to see that the reviewer finds our work: (a) resembling everyday physical situations, (b) detailed in terms of diagnostics, and (c) facilitator of large-scale and rapid model comparisons.
>
> **Q: Difference with VideoPhy**
>
> We strongly disagree with the reviewer’s opinion about the limited novelty of our work. We highlight that VideoPhy2 is a non-trivial improvement over VideoPhy:
> - At its core, VideoPhy focuses on evaluating the physical commonsense and focuses on interactions between diverse states of matter (solid-solid, solid-fluid, and fluid-fluid) on much simpler prompts that do not contain multiple events. On the other hand, VideoPhy2 (ours) evaluates an entirely different capability where the focus on real-world actions (i.e., object interactions as well as physical activities) containing multiple events in more complex (longer) prompts. As a result, there is a huge methodological difference in their data collection process between the two work (Section 2 of VideoPhy vs Section 2 of VideoPhy2).
> - The coverage of VideoPhy2 (3940) is much higher (6x) than VideoPhy (344 prompts). In addition, the scale of the annotations is much more fine-grained (1-5) for semantic adherence (SA) and physical commonsense (PC) in VideoPhy2 in comparison to VideoPhy (binary decisions). In Line 248-252, we highlight that VideoPhy suffers from inherent evaluation bias because the human annotators can “see” the prompt while making PC decisions but VideoPhy2 runs separate evaluation for SA and PC (Figure Figure 17/18).
> - VideoPhy2 is much more challenging than VideoPhy. The best performing model in VideoPhy paper, CogVideoX-5B, achieves a very low score on VideoPhy2 i.e., 25%. Further, VideoPhy2 contains a hard subset of actions that are more challenging for stronger models like Wan2.2 whereas there is no such subset in VideoPhy.
> - VideoPhy2 introduces an entirely new feature of physical rule annotation that is absent in VideoPhy. This physical rule annotation is non-trivial to acquire, hence, we created a new pipeline to get its judgment. In particular, we first caption each video with a VLM and use LLM to get candidate physical rules. Then, we ask human annotators to make their judgements about the candidate's physical rules and manually write any new rule that is missing in the automatic process. While VideoPhy uses cherry-picked examples to get mode of failures, these physical rules allow a quantitative way to assess the video model failures which is much more precise.
> - In table 4 and 5, we show that the automatic evaluator trained in VideoPhy2 is much more powerful than the automatic evaluator from VideoPhy work. Infact, the gaps are non-trivially large (upto 50% relative improvements). Hence, we not only improved the quality of the dataset but also the autoeval.
>
> We will add this discussion in the camera-ready version to distinguish with VideoPhy more clearly.
>
>
> **Q: Comparison with cutting edge models**
>
> - Firstly, we clarify that the Table 4 shows that our autoeval achieves the best performance in comparison to a large set of models on semantic adherence and physical commonsense judgment. The second best baseline was VideoCon-Physics. That is why, we compare our model with the second best model on joint score metric (Table 5).
> - Secondly, we highlight that Gemini is a frontier and proprietary model that is at par or even better than GPT models. In our experiments (Table 4), we highlight that our autoeval is much better than Gemini. This is attributed to the lack of synthetic videos and physical commonsense understanding in Gemini which is inducted into our model through high-quality human eval data.
> - To further address the reviewers comment, we got semantic adherence (SA) and physical commonsense (PC) judgments from an upgraded version i.e., Gemini-2.5-Flash. We provide the correlation between the ground-truth SA and PC with the predictions below:
> | Method                                       | Avg   | SA     | PC    |
> |----------------------------------------------|-------|--------|-------|
> | Gemini-Flash-2                               | 18.5  | 26     | 11    |
> | Gemini-Flash-2.5                             | 20.5  | 31     | 10    |
> | VideoPhy2-AutoEval                           | 42    | 47     | 37    |
> | Rel. Improvement in comparison to Gemini-2.5 | **+104%** | **+51.6%** | **+270%** |
>
> - Interestingly, we find that VideoPhy2-AutoEval performs much better than Gemini-Flash-2.5, consistent with our findings for Gemini-Flash-2.0. This suggests that recent improvements in state-of-the-art VLMs are not focused on enhancing semantic adherence or physical commonsense judgments for synthetic videos. Therefore, using our open and free AutoEval remains the more reliable choice for the future.

---

> > ### Author Response · Authors · 2025-11-16
> > **Author Rebuttal 2**
> >
> > **Q: VideoPhy2 to improve physical commonsense of video models**
> >
> > - We respectfully highlight that such experiments are far beyond the scope of our “datasets and benchmarks track” (selected as the primary area) paper. The purpose of an evaluation paper is to provide a high-quality dataset that supports assessing model capabilities, evaluations, and analysis. As the reviewer noted in the strengths, we have achieved many of these goals. - Developing new methods to address the identified gaps would constitute an entirely separate paper or project.
> > - Our primary goal in this work is to assess whether modern video models can generate physically accurate videos across a diverse range of real-world actions, a gap that has not been systematically examined before.
> > - To achieve this, we designed a complete data collection and evaluation framework. This includes identifying real-world actions that require physical commonsense reasoning, generating large-scale captions using LLMs, and conducting extensive human evaluations. We also introduce a challenging subset of actions (e.g., hula-hooping, nunchuck spinning) that we show are significantly more difficult than typical cases and expose clear weaknesses in existing models.
> > - Furthermore, we are among the first to curate both semantic adherence and detailed physical rules to quantify physical commonsense. This enables rigorous analysis of model failure modes rather than relying on anecdotal examples. We also train an automatic evaluator on our annotations and demonstrate that it generalizes well to unseen prompts and videos. Importantly, our approach outperforms proprietary and often costly baselines such as Gemini.
> > - Looking ahead, using insights from VideoPhy-2 to improve video generation is a promising direction for model builders. In particular, VideoPhy2-AutoEval can serve as a reward model that provides feedback on generated content, enabling refinement of video models through reinforcement learning or alignment-based approaches [1, 2, 3].
> >
> > [1] Think Before You Diffuse: https://arxiv.org/abs/2505.21653
> >
> > [2] Video Prediction Models as Rewards for Reinforcement Learning: https://arxiv.org/abs/2305.14343
> >
> > [3] Video Diffusion Alignment via Reward Gradients: https://arxiv.org/abs/2407.08737

---

> > > ### Comment · Reviewer_74jg · 2025-11-28
> > >
> > > The author's reply resolved my question, so I chose to increase my score.

---

> > > > ### Author Response · Authors · 2025-11-28
> > > > **Score Change Reminder**
> > > >
> > > > Hi Reviewer 74jg,
> > > >
> > > > We are glad to hear that our response addressed your questions. You mentioned that you have decided to increase the score, but we still see your original rating on our end. Could you please update the score if this is an error on your side?

---

### Official Review · Reviewer_VSb4 · 2025-10-31

**Soundness:** 2
**Presentation:** 1
**Contribution:** 2
**Rating:** 4
**Confidence:** 3

**Summary:**

The paper introduces VIDEOPHY-2, a large-scale benchmark for testing whether text-to-video generation models can follow basic physical commonsense. It expands the earlier VIDEOPHY dataset with 197 actions and 3,940 detailed prompts, focusing on action-centric, physics-rich scenarios. Authors also train VideoPhy-2-AutoEval, an automatic evaluator for fast assessment on their proposed dataset.

**Strengths:**

1. The paper proposes a large-scale, carefully annotated benchmark for assessing the physical understanding ability of video generation models.
2. An automatic evaluator is trained to approximate human judgments, which is useful in principle, though not always effective in practice.

**Weaknesses:**

1. I really dislike the presentation. Figure 2 does not clearly correspond to Section 2 (for example, the dense captioning component is not reflected in the figure). In Section 3.1, the distinction between PC and PR is unclear. Several parts are verbose and poorly organized; for instance, around line 275, experimental descriptions appear within the evaluation methodology section. Similar issues occur throughout the paper.
2. The experimental setup is unfair, as some models use upsampled captions while others do not. The authors should generate and evaluate videos separately for fair comparison.
3. The experimental result and tables should be more detailed. The main results table (Table 2) should explicitly report all metrics such as SA and PC.
4. According to Table 4, the pearson's correlation between VIDEOPHY-2-AUTOEVAL and human ratings is below 0.5, indicating that the proposed automatic evaluator does not generalize well to unseen scenarios. Therefore, I have concerns about the benchmark’s reliability for automatic evaluation, especially since human evaluation of video generation models is labor-intensive and inefficient.

**Questions:**

1.See Weaknesses.
2. What is the inter-annotator agreement among human evaluators? The paper should include an analysis of annotation consistency to verify the reliability of human evaluation.
3. The amount of work in this paper is impressive, and I appreciate the effort. However, the overall writing quality is poor. The authors should seriously reconsider how to present and organize their work more clearly and coherently.

---

> ### Author Response · Authors · 2025-11-17
> **Author Rebuttal**
>
> We thank the reviewer for their comments.
>
> **Q: Presentation**
>
> - **Section 2:** We respectfully disagree with the reviewer’s opinion that Figure 2 does not correspond to Section 2. The original LLM generated prompt (shown in the Figure 2) has the identical semantics as its dense captioning version. The dense captioning is required for several video models because it helps in setting up with the peripheral knowledge better (e.g., including details about lighting, camera angles). The human workers are still shown the original prompt (Figure 2) in the annotation process (Line 253-255). Thus, the Figure clearly reflects our design choices. We provide example prompts and its denser version in Appendix Table 12 (also mentioned in Line 160-161). Due to the density of the long captions, they won’t fit well in the Figure 2 and don’t do justice to other components of the entire pipeline. To avoid any future confusion, we will explicitly highlight the dense captioning step for video generation in Figure caption.
> - **PC vs PR:** As mentioned in Line 73-75 and Section 3.1, physical commonsense [1] is simply about the physical likelihood of the generated video (V), an assessment that can often be made by humans by relying on their real-world experience. On the other hand, physical rule judgment is whether a physical rule (R) followed in the generated video (V). According to Figure 2, “The ball moves through the air in an arc form” is followed in the generated video while “The tennis ball changes position after being hit” is not followed. The purpose of physical rule judgments is to get a sense of fine-grained signals used by human workers to make their physical commonsense decisions (Line 252-258).
> - **Line 275**: We thank the reviewer for pointing this out. We clarify that we recall some early experimental results to train our own automatic evaluator instead of using existing ones. We understand that it might be confusing for some readers. As a result, we will fix the writing style in a way where experimental related phrasings are used in the setup and experiments section.
>
> [1] VideoPhy (ICLR 2025): https://arxiv.org/abs/2406.03520
>
> **Q: Upsampled captions**
>
> - We believe there is a misunderstanding regarding the role of upsampled prompts. As shown in Table 12, the upsampled prompts preserve the same semantic meaning as the original shorter prompts. Their purpose is to provide expanded context that helps video models render higher-quality videos.
> - Older models such as Hunyuan-13B and VideoCrafter2 are constrained by the CLIP text encoder’s 77-token limit, which prevents them from accepting long, dense captions. In contrast, the upsampled prompt feature is available only in more recent video models, and these models rely on longer prompts to perform well. When evaluated with short prompts, the newer models produce noticeably lower-quality videos. For example, Cosmos-7B performs poorly with original prompts but improves significantly when given upsampled prompts, which is also the default method recommended by the model creators.
> - Our design choice ensures that every model is evaluated using its best possible prompt format. We do not want to penalize strong video models by forcing them to use short prompts when they are trained for and dependent on longer prompts. This leads to a fairer comparison and better reflects real-world usage.

---

> > ### Author Response · Authors · 2025-11-17
> > **Rebuttal continued**
> >
> > **Q: Reporting SA and PC**
> >
> > We thank the reviewer for this feedback. We already present the distribution of semantic adherence and physical commonsense scores in Figure 8 and Figure 9 (Appendix N). We will highlight this in the main text too.
> > To further address reviewer’s comment, we will add the SA and PC scores for Table 3 results as shown below:
> > | SA              | All | Hard | Physical Activities | Object Interactions |
> > |-----------------|-----|------|---------------------|---------------------|
> > | Wan2.2-27B-A14B | 3.8 | 3.7  | 3.8                 | 4.0                 |
> > | Wan2.1-14B      | 3.3 | 3.1  | 3.3                 | 3.5                 |
> > | CogVideoX-5B    | 3.0 | 2.4  | 3.0                 | 3.0                 |
> > | Cosmos-Diff-7B  | 3.2 | 2.8  | 3.2                 | 3.3                 |
> > | Hunyuan-13B     | 2.9 | 2.7  | 2.9                 | 2.9                 |
> > | VideoCrafter-2  | 2.6 | 2.4  | 2.6                 | 2.6                 |
> > | SVD-I2V         | 2.1 | 2.0  | 1.9                 | 2.4                 |
> > | Ray2            | 3.1 | 2.8  | 3.1                 | 3.1                 |
> > | Sora            | 2.8 | 2.3  | 2.7                 | 3.0                 |
> >
> > | PC              | All | Hard | Physical Activities | Object Interactions |
> > |-----------------|-----|------|---------------------|---------------------|
> > | Wan2.2-27B-A14B | 4.1 | 4.0  | 4.1                 | 4.0                 |
> > | Wan2.1-14B      | 3.7 | 3.4  | 3.6                 | 3.7                 |
> > | CogVideoX-5B    | 3.7 | 3.3  | 3.6                 | 3.9                 |
> > | Cosmos-Diff-7B  | 3.5 | 3.2  | 3.5                 | 3.5                 |
> > | Hunyuan-13B     | 3.5 | 3.3  | 3.5                 | 3.7                 |
> > | VideoCrafter-2  | 3.1 | 2.9  | 3.0                 | 3.3                 |
> > | SVD-I2V         | 3.1 | 3.0  | 3.2                 | 3.1                 |
> > | Ray2            | 3.4 | 3.2  | 3.3                 | 3.5                 |
> > | Sora            | 3.6 | 3.5  | 3.6                 | 3.6                 |
> >
> > **Q: Automatic Evaluation**
> >
> > - We agree that the automatic evaluation scores show only moderate correlation with human ratings. However, it is important to view these numbers in context: VideoPhy2-AutoEval substantially outperforms all baselines, including costly proprietary models like Gemini (up to 250% relative improvement) and prior video understanding models such as VideoCon-Physics (up to 50% relative improvement). These results demonstrate that our method offers a significant advance in automatically assessing physical commonsense in generated videos.
> > - We do not claim that our evaluator fully solves automatic physical commonsense assessment. Instead, we position it as a meaningful step toward more scalable and accurate evaluation. With an industrial-scale budget, one could collect even larger human-annotated datasets using our framework, likely enabling further gains on our test set.
> > - In practice, our automatic evaluator provides the most reliable signal available today compared to existing video understanding models. That said, we agree that human evaluation remains essential for final validation, particularly for deployment settings.
> > - We also emphasize that most industrial labs have the resources to conduct robust human evaluations, and our benchmark provides a clear protocol for evaluating physical commonsense in real-world action videos.
> > - Overall, physical commonsense is a challenging and underexplored problem. We are among the few works tackling it in realistic settings, and we intentionally avoid simplifying the task merely because current models struggle with it. Our benchmark aims to reflect real-world difficulty while pushing the field forward.
> >
> > **Q: Human Evaluation**
> >
> > - Despite the challenging nature of the tasks, we take several measures to ensure that human evaluation is high-quality and consistent.
> > - As mentioned in Line 203-208, all annotators underwent structured training guided by a detailed rubric with clear examples, ranging from “very unlikely” (1) to “very likely” (5), to anchor their judgments and establish a shared understanding of the scale. To further reduce individual bias and capture a stable consensus, each video was evaluated by three annotators. This process yielded a high inter-annotator agreement of 80% (comparable to agreement scores reported in prior work [1]), confirming the consistency and validity of our framework.
> >
> > [1] VideoPhy (ICLR 2025): https://arxiv.org/abs/2406.03520

---

> > > ### Author Response · Authors · 2025-11-28
> > > **Reviewer Reminder**
> > >
> > > Hi Reviewer VSb4,
> > >
> > > Thank you again for handling our paper. Since the author–reviewer discussion period is coming to an end, could you please let us know if there are any further questions?

---

### Official Review · Reviewer_QRAy · 2025-10-31

**Soundness:** 3
**Presentation:** 3
**Contribution:** 3
**Rating:** 6
**Confidence:** 4

**Summary:**

This paper introduces VIDEOPHY-2, a large-scale benchmark for evaluating physical commonsense in text-to-video generation. The dataset consists of 3,940 LLM-generated prompts covering 197 real-world actions, with detailed human annotations on semantic adherence, physical commonsense, and rule violations. It includes a hard subset emphasizing physics-rich and multi-event scenarios (e.g., backflip, throwing discus). The authors also develop VIDEOPHY-2-AUTOEVAL, a fine-tuned video-language model trained on 50K human annotations for scalable automatic evaluation. Extensive experiments on leading open and closed video models (e.g., Wan2.2-27B, Sora, CogVideoX) show that current models struggle with physical plausibility—particularly with conservation laws of mass and momentum—achieving only 47.7% joint adherence on the hard subset.

**Strengths:**

1. The proposed benchmark addresses an important gap in evaluating physical realism in video generation, a key step toward world-modeling AI, it extends the scale with more fine-grained physical law annotations compared to previous version VIDEOPHY.

2. The proposed VIDEOPHY-2-AUTOEVAL is valuable and achieves large correlation gains over strong baselines (e.g., Gemini-2.0-Flash-Exp).

3. Thorough experiments and insightful analysis: The fine-grained breakdown of violated physical laws (e.g., momentum, elasticity) provides deep diagnostic insight.

**Weaknesses:**

1. Missing related work discussion: The paper does not reference Impossible Videos, which also proposes a benchmark including evaluating violations of physical laws in video generation. A detailed comparison and discussion are needed to clarify the conceptual and methodological differences between VIDEOPHY-2 and that work.

2. Prompt ambiguity in multi-event scenarios: The authors encourage LLMs to generate prompts depicting multiple events within a single description. While this increases task difficulty, it also introduces ambiguity when evaluating semantic adherence—for instance, when a video generator accurately follows only part of the prompt. How is semantic adherence rated in such cases? Is there a standardized or clearly defined evaluation guideline to handle partial adherence?

**Questions:**

1. Does training on VIDEOPHY-2 improve a model’s physical commonsense generation?
2. Could the authors release example failure cases and annotation guidelines to help the community standardize future evaluation protocols?

---

> ### Author Response · Authors · 2025-11-16
> **Author Rebuttal**
>
> We thank the reviewer for their encouraging feedback. We are motivated to see that the reviewer finds our work: (a) addressing an important gap in evaluating the physical realism, (b) valuable that achieves state-of-the-art performance, and (c) insightful with deep diagnostics.
>
> **Q: Discussion on Impossible Videos**
>
> - We thank the reviewer for highlighting this work. We note that our contributions and story is quite different from this work. Specifically, Impossible Videos (IPV) aims to evaluate the ability of the video generation models to create “anti or counterfactual real-world” videos (e.g., A person pours milk into a glass cup half filled with milk, but the amount of milk in the glass cup does not change at all). In contrast, our work focuses on testing the ability of the models to follow physical commonsense and semantic adherence for “real-world” actions. In this regard, the goals of these works are entirely different.
> - Methodologically, IPV collects 260 prompts while we cover a much larger set of 4000 prompts. While IPV just evaluates “visual quality” and “semantic adherence” as Yes/No judgment from human annotators, our work performs much more fine-grained and broader evaluation. For instance, we assess the physical commonsense and semantic adherence on a scale of 1-5 which captures finer human preferences. Further, we annotate for the physical rules that are not followed in each video.
> - In addition, we train an automatic evaluator for scalable evaluation on our dataset while IPV just tests the ability of existing VideoLLMs to solve their task instead of training their own. As a result, the practitioners will have to rely on expensive proprietary models (GPT-4o) to do automatic evaluation on IPV but our AutoEval is open and free.
> - In terms of our findings, it is surprising that even the best models are not good at capturing the physics for “real-world actions”. In this regard, our work encourages the model builders to first fix the lack of in-distribution instead of far out-of-distribution (i.e., anti real-world) capabilities as a first goal.
> - We will add this discussion in the related work section of the camera-ready version.
>
> **Q: Clarification on semantic adherence**
>
> - We thank the reviewer for this pertinent question. We agree that the task for semantic adherence is inherently subjective with the possibility of partial adherence when multi-events are present in the prompt (e.g., rolling ball then stopping). However, we note that the partial adherence issue is not limited to multi-events prompts. Even simplistic prompts (e.g., a ball bouncing on the floor) may show partial adherence. This shows that the task of semantic adherence (or physical commonsense) are subjective tasks.
> - To alleviate this issue, we ask three human annotators to judge each video to get an average sense of adherence. Further, we provide very detailed instructions to the annotators before they perform the task annotations. Figure 12 and 13 of the supplementary contains the definitions provided to human annotators for each category. We will mention it here -
>
> **(5) Perfect:** The video fully and accurately adheres to ALL aspects of the prompt, including subtle details, mood, and style.
>
> **(4) Good:** Most elements of the prompt are depicted correctly, with only minor omissions or deviations.
>
> **(3) Partial:** Some key elements are correct, but there are significant inconsistencies or omissions that impact the core request.
>
> **(2) Poor:** Many key aspects are missing or clearly wrong. The video barely resembles the prompt.
>
> **(1) No Alignment:** The video does not match the prompt at all.
>
> **Q: VideoPhy2 to improve physical commonsense of video models**
>
> - We respectfully note that this question extends beyond the scope of our current paper. Our primary goal in this work is to assess whether modern video models can generate physically accurate videos across a diverse range of real-world actions, a gap that has not been systematically examined before.
> - To achieve this, we designed a complete data collection and evaluation framework. This includes identifying real-world actions that require physical commonsense reasoning, generating large-scale captions using LLMs, and conducting extensive human evaluations. We additionally introduce a challenging subset of actions (e.g., hula-hooping, nunchuck spinning) that we show are significantly more difficult than typical cases and reveal clear weaknesses in existing models.
> - Furthermore, we are among the first to curate both semantic adherence and detailed physical rules to quantify physical commonsense. This enables rigorous analysis of model failure modes rather than relying on anecdotal examples. We also train an automatic evaluator on our annotations and demonstrate that it generalizes well to unseen prompts and videos. Importantly, our approach outperforms proprietary (and often costly) baselines such as Gemini.

---

> ### Author Response · Authors · 2025-11-16
> **Rebuttal continued**
>
> - Looking forward, using insights from VideoPhy-2 to improve video generation is a promising direction for model builders. In particular, VideoPhy2-AutoEval can serve as a reward model that provides feedback on generated content, enabling refinement of video models through reinforcement learning or alignment approaches [1,2,3].
>
> [1] Video Prediction Models as Rewards for Reinforcement Learning: https://arxiv.org/abs/2305.14343
>
> [2] Video Diffusion Alignment via Reward Gradients: https://arxiv.org/abs/2407.08737
>
> [3] Think Before You Diffuse: https://arxiv.org/abs/2505.21653
>
> Q: Annotation guidelines and failure cases
>
> - We have already released the annotation guidelines (Figure 11/12/13) and human interface (Figure 17/18) in the supplementary material.
> - In addition, we will release the entire dataset, auto eval, and code in the camera-ready version.

---

> > ### Comment · Reviewer_QRAy · 2025-11-27
> >
> > Thanks for the authors' response. I have no further concerns.

---

> ### Author Response · Authors · 2025-11-28
> **Author Response [Edited]**
>
> ```Re: [URGENT ICLR 2026] Response to leaked reviewer/AC identities```
>
> Hi AC,
> We understand that the reviewer will not be able to engage further on this thread. We would like to highlight that the reviewer did not raise any additional questions after our rebuttal and appeared to be leaning positively toward our submission.
>
> ```Original comment:```
>
> Hi Reviewer QRAy,
>
> Thank you for your reply. We are glad to hear that our rebuttal addressed all your questions. Please let us know if anything else needs clarification. If there are no further concerns, we would be grateful if you could consider raising the score, as it would meaningfully help improve the chances of acceptance and benefit the community.

---

### Meta-Review · Area_Chair_xGYD · 2026-01-12

**Summary:**

This paper introduces an action-centric dataset for evaluating physical commonsense in video generation. This paper received two positive scores before rebuttal and three (Reviewer 74jg increases score) after rebuttal. Most concerns are addressed by the rebuttal, and the remains are not critical. This new benchmark should benefit the community. I suggest accepting the paper.

**Reviewer Concerns:**

Most concerns from Reviewer QRAy, 74jg, and tmU3 are addressed.
Reviewer VSb4 points out his concern about writing, experiment settings, and the reliability of automatic evaluation. The first two are addressed, but the reliability of automatic evaluation remains unconvincing.

**Reviewer Scores:**

This paper receives two positive scores before rebuttal and three (Reviewer 74jg increases score) after rebuttal. Reviewer VSb4 may keep the original score since his concerns are not well addressed.

---

### Decision · Program_Chairs · 2026-01-26

Accept (Poster)